



# The control of climate sensitivity on variability and change of summer runoff from two glacierised Himalayan catchments

Sourav Laha[1], Argha Banerjee[1], Ajit Singh[2], Parmanand Sharma[2], and Meloth Thamban[2]

[1]Earth and Climate Science, Indian Institute of Science Education and Research (IISER) Pune, Pune-411008, India
[2]National Centre for Polar and Ocean Research (NCPOR), Ministry of Earth Sciences, Vasco-da-Gama, Goa-403804, India

**Correspondence:** Argha Banerjee (argha@iiserpune.ac.in)

**Abstract.** The response of catchment runoff to climate forcing is determined by its climate sensitivity. We investigate the sensitivity of summer runoff to precipitation and temperature changes in winter-snow dominated Chandra (western Himalaya), and summer-rain dominated upper Dudhkoshi (eastern Himalaya) catchments in order to understand the nature of climate-change impact on the mean summer runoff and its variability. The runoff over the period 1980–2018 is simulated with a

semi-distribute hydrologic model, which is calibrated using available discharge and glacier mass loss data. An analysis of the interannual variability of the simulated summer runoff reveals that the runoff from the glacierised parts of the catchments is sensitive to temperature changes, but is insensitive to precipitation changes. The behaviour of the summer runoff from the non-glacierised parts is exactly opposite. Such precipitation-independent runoff from the glacierised parts stabilises the catchment runoff against precipitation variability to some degree. With shrinking glacier cover over the coming decades, the

summer runoff from the two catchments is expected become more sensitive to the precipitation forcing and less sensitive to the temperature forcing. Because of these competing effects, the impact of the glacier loss on the interannual variability of summer runoff may not be significant. However, the characteristic 'peak water' in the long-term mean summer runoff, which is caused by the excess meltwater released by the shrinking ice reserve, may lead to a detectable signal over the background interannual variability of runoff in these two catchments.

## 1 Introduction

The glacier-melt component in the Himalayan water cycle buffers the river basins against drought (Pohl et al., 2017; Pritchard, 2019), but renders them susceptible to climate-change impacts (Azam et al., 2021). The ongoing changes in temperature and precipitation (Hugonnet et al., 2021) are causing Himalayan glaciers to shrink. The glacier loss is expected to continue, deplet-

ing the Himalayan ice reserve considerably by 2100 (Kraaijenbrink et al., 2017). While the recent glacier loss is contributing excess water to the Himalayan rivers, a decline in the glacier contribution to runoff by the end of this century is likely as the ice reserve shrinks (Huss and Hock, 2018). This 'peak water' is a typical characteristic of the long-term response glaciersied





catchments to sustained warming (Hock et al., 2005). Due to a lack of long-term glacio-hydro-meteorological data from the region (Miller et al., 2012; Azam et al., 2021), and uncertain climate projections (Joseph et al., 2018), a satisfactory under-

standing of the future changes in the runoff from Himalayan catchments may still be lacking. In this context, an understanding of the climate sensitivity of runoff from these catchments may provide useful clues about the future change in catchment runoff and its variability in the high Himalaya, and may also helpful in understanding the nature of climate response of glacierised catchments in general.

Climate sensitivity of runoff is defined as the change in runoff due to a unit perturbation in a forcing variable e.g., precipita-

tion or temperature (e.g. Zheng et al., 2009). These sensitivities can be obtained simply by regressing the observed variability of runoff to those of its meteorological drivers (e.g. Duell, 1994; Berghuijs et al., 2014). Here the perturbations considered are the physically-consistent natural variation of the forcing parameters. This method may be preferable over the commonly adopted strategy of computing the runoff response to an ad-hoc shift to a meteorological parameter in order to constrain the sensitivity. Long-term runoff changes due to temperature and precipitation forcing had previously been studied using climate-sensitivity

based approach for both glacierised (Chen and Ohmura., 1990) and non-glacierised catchments (Dooge et al., 1999; Zheng et al., 2009; Vano et al., 2012). Climate-sensitivity based predictions for future changes in runoff are reliable as long as the predicted changes lie in the range of the interannual variability of runoff over the calibration period. An related linear-response approach has been used (Chen and Ohmura., 1990) to study the so-called 'glacier-compensation effect' – a systematic non-monotonic dependence of the coefficient of variation of runoff across catchments on the corresponding fractional glacier cover

with a minimum at a moderate glacier cover .

The climate sensitivity of the runoff from high Himalayan catchments remains poorly constrained in the literature. This gap motivates the present study of the sensitivity of summer runoff of two glacierised Himalayan catchments, Chandra (western Himalaya) and upper Dudhkoshi (eastern Himalaya), to temperature and precipitation forcing. The Variable Infiltration Capacity (VIC) model (Liang et al., 1996), augmented with a glacier-melt module that assumes a static glaciers cover, is used to simulate

summer runoff over the period 1980–2018. The simulated time series is analysed to obtain the sensitivity of summer runoff to annual precipitation and summer temperature. The sensitivities of the summer runoff from the glacierised and non-glacierised parts of the catchments are also analysed. The sensitivities obtained are then used to investigate the nature of possible changes in the mean and the variability of summer runoff of the catchments as the glacier cover shrinks due to a warming climate. We also attempt to explain the glacier-compensation effect, and to estimate the timing and magnitude of 'peak water' using the

climate-sensitivities.

## 2   Study area

We considered two high Himalayan catchments with contrasting climate regimes: winter snow dominated Chandra (Indus basin, the western Himalaya), and summer rain dominated upper Dudhkoshi (Ganga basin, the eastern Himalaya) (Figs. 1a–1c). The Chandra catchment is in Lahaul-Spiti district, Himachal Pradesh, India. The upper Dudhkoshi catchment is in Solukhumbu

district of Nepal. About 50% of the annual precipitation in Chandra catchment occurs during the winter months (Fig. 1e) due to

**Figure 1.** Maps of (a) Chandra and (b) upper Dudhkoshi catchments with the glaciers (Cyan polygons) and the stream network (purple lines). The red solid circles (triangles) are the meteorological (hydrological) stations. The ERA5 grid boxes are shown with solid gray lines in the background. c) A location map of Chandra (red solid triangle) and upper Dudhkoshi (blue solid triangle) catchments on a grey-scale elevation map (Amante et al., 2009). In the remaining plots red (blue) symbols/lines refer to Chandra (upper Dudhkoshi) catchment. d) Area-elevation distribution of the catchments (solid lines + solid symbols), and the corresponding glacierised parts (dashed lines + solid symbols). e) Mean monthly precipitation (solid lines + solid symbols), along with the monthly snowfall (dashed lines + solid symbols). d) Mean monthly temperature profiles (solid lines + solid symbols). **3**





**Table 1.** A summary of the characteristics of Chandra and upper Dudhkoshi catchments. The meteorological variables are from bias-corrected reanalysis data (Hersbach et al., 2020), the hydrological data are from model simulations (the present study). The glacier mass-balance and area-loss estimates are from the existing literature (supplementary Table S3).

| Catchment | Chandra | Upper Dudhkoshi |
|---|---|---|
| Basin | Indus | Ganga |
| Area (km$^2$) | 2440 | 1190 |
| Outlet | Tandi | Phadking |
| Elevation range (m a.s.l) | 2850–6500 | 2600–7900 |
| Glacierised fraction | 0.25 | 0.20 |
| Annual temperature (°C) | −5.5 | −4.7 |
| Annual precipitation (mm yr$^{-1}$) | 1610 | 1531 |
| Summer precipitation / winter precipitation | 0.5 | 6.8 |
| Liquid precipitation/ solid precipitation | 0.5 | 9.7 |
| Glacier area loss (% decade$^{-1}$) | 1.1–5.5 | 1.2–4.2 |
| Glacier mass balance (m w.e. yr$^{-1}$) | $-0.13 \pm 0.11$ to $-0.56\pm0.38$ | $-0.26 \pm 0.13$ to $-0.52\pm0.22$ |
| Annual runoff (m yr$^{-1}$) | 1.25 | 0.99 |

the western disturbances (Bookhagen and Burbank, 2010), and the influence of Indian summer monsoon is relatively weak. In upper Dushkoshi catchment, more than 80% of the precipitation happens during the summer months (Fig. 1e) with a dominant influence of Indian summer monsoon. Here, and in the rest of this paper, summer refers to the high-discharge period from May to September (Azam et al., 2019). Due to the contrasting seasonality of precipitation, the ratio of solid to liquid precipitation in

Chandra and upper Dudhkoshi catchments are 0.5 and 9.7, respectively. Apart from the above differences in the precipitation regimes, the two catchments are quite similar in terms of the mean annual temperature, catchment hypsometry, elevation range, specific summer runoff, glacier fraction, and the recent rates of glacier loss (Figs. 1d–1f, and Table 1).

## 3   Data and methods

We simulated river runoff of the above two catchments over the period 1980–2018 using VIC model (Liang et al., 1996). The

model was forced with bias-corrected reanalysis data (Hersbach et al., 2020), and was augmented with a temperature-index based glacier-melt module (Hock, 2003). The glacier-melt module assumed a static glacier cover as the observed change in glacier area was relattively small over the simulation period (Table 1). The model was calibrated with the help of available summer runoff data and geodetic glacier mass balance from both the catchments. The simulated runoff was then utilised to obtain the sensitivities of summer runoff to summer temperature and annual precipitation changes. Note that throughout the

paper, the annual quantities correspond to the hydrologic year: from 1st October of a calendar year to 30th September of the next (e.g. Azam et al., 2019). Below, we first present our climate-sensitivity based theoretical approach towards understanding the variability and change in runoff. Subsequently, we discuss the details of input data, model simulations, and model calibration.



### 3.1 Climate sensitivity of runoff

The climate sensitivity of summer runoff $Q$ is defined as the change in runoff due to a unit perturbation in a meteorological
forcing parameter (e.g. Zheng et al., 2009). Here, we concentrate of the sensitivity of the specific summer runoff $(Q)$ from the
studied catchments to changes in annual precipitation $(P)$ and mean summer temperature $(T)$. Note that summer runoff and
annual precipitation are in m yr$^{-1}$, and mean summer temperature is in °C. Hereinafter, for any given variable $X$, we use the
symbols $X_0, \delta X, \Delta X$, and $\sigma_X$ to denote the long-term mean, the anomaly for a given year, the change in the long-term mean,
and the standard deviation, respectively.

### 3.1.1   Estimating the climate sensitivities

The sensitivities of summer runoff can be used (e.g. Zheng et al., 2009) to relate the anomalies of summer runoff $(\delta Q)$, annual
precipitation $(\delta P)$, and summer air-temperature $(\delta T)$ as follows.

$$\delta Q = s_P \delta P + s_T \delta T. \tag{1}$$

Here, precipitation sensitivity is denoted by $s_P \doteq \frac{\partial Q}{\partial P} = \frac{\partial \delta Q}{\partial P}$, and temperature sensitivity is denoted by $s_T \doteq \frac{\partial Q}{\partial T} = \frac{\partial \delta Q}{\partial T}$. In
Eq. (1), a possible bilinear interaction term proportional to $\delta T \delta P$ (Lang., 1986) was not considered. We confirmed this correc-
tion term, when included in the regression for the catchment studied, was not significant.

In order to estimate the sensitivity coefficients $s_T$ and $s_P$, we regressed simulated time series of $\delta Q$ for the catchments
during 1997–2018 with the corresponding time series of $\delta T$ and $\delta P$. The standard error of the fits obtained the corresponding
uncertainties. The sensitivities estimated from the simulated $\delta Q$ over 1997–2018 were validated using the simulated variability
during 1980–1996, with the help of metrics Nash-Sutcliffe efficiency (NSE) and root mean squared error (RMSE).

We also considered the runoff from glacierised part of the catchments $Q^{(g)} \doteq Q_0^{(g)} + \delta Q^{(g)}$, and that from the non-glacierised
part of the catchments $Q^{(r)} \doteq Q_0^{(r)} + \delta Q^{(r)}$. The corresponding sensitivities were defined in a similar way,

$$\delta Q^{(g)} = s_P^{(g)} \delta P + s_T^{(g)} \delta T, \tag{2}$$
$$\delta Q^{(r)} = s_P^{(r)} \delta P + s_T^{(r)} \delta T. \tag{3}$$

These climate sensitivity coefficients and the corresponding uncertainties were estimated using the simulated annual anomalies
$\delta Q^{(g)}$ and $\delta Q^{(r)}$, and the corresponding $P$ and $T$ anomalies over the period 1997–2018. With glacier fraction denoted by $x$, the
following relations connect the quantities defined for the glacierised and non-glacierised part of the catchments to onesdefined
for the whole catchment.

$$\delta Q = x \delta Q^{(g)} + (1-x) \delta Q^{(r)}, \tag{4}$$
$$s_T = x s_T^{(g)} + (1-x) s_T^{(r)}, \tag{5}$$
$$s_P = x s_P^{(g)} + (1-x) s_P^{(r)}. \tag{6}$$

Apart from the sensitivities of summer runoff, we also computed the precipitation and temperature sensitivities of glacier
mass balance using the corresponding simulated interannual variability. For the precipitation sensitivity of glacier mass bal-





ance, we defined it to be the mass-balance change due to a 10% change in precipitation for the ease of comparison with the
corresponding values available in the literature (e.g. Mölg et al., 2012).

### 3.1.2    Variability of summer runoff and its changes

The climate sensitivities defined above determine the interannual variability of summer runoff given those of $P$ and $T$ as,

$$\sigma_Q = \sqrt{s_T^2\sigma_T^2 + s_P^2\sigma_P^2}, \tag{7}$$

where $\sigma_Q, \sigma_P$, and $\sigma_T$ are standard deviations of $Q, P$ and $T$, respectively. An implicit assumption here is that $\delta P$ and $\delta T$ are
uncorrelated over the simulation period, which we verified to be true at $p < 0.05$ level. Equation (7), together with Eqs. (5)
and (6), can be used to explain the first order differences in $\sigma_Q$ between different catchments in terms of the differences in
the corresponding climate sensitivities, glacier fraction $(x)$ , climate variability $(\sigma_T$, and $\sigma_P)$. The first order changes in $\sigma_Q$
under a changing climate can also be estimated using the above equation using projected values of $\sigma_T$, and $\sigma_P$. For the studied
catchments, we used this relation to investigate the impact of changing glacier cover on the future interannual variability of
summer runoff $\sigma_Q$.

### 3.1.3    Multidecadal changes in mean summer runoff

The climate sensitivity coefficients defined above can also be used to predict changes in summer runoff $(\Delta Q)$ over the next few
decades for any given changes annual precipitation $(\Delta P)$ and mean summer temperature $(\Delta T)$. For glacierised catchments,
apart from changing $P$ and $T$, the changes in glacier fraction $x$ will also influence the future runoff. Defining time-varying
glacier fraction $x \doteq x_0 + \Delta x$, the following linear-response equation can be constructed,

$$\Delta Q \;\;=\;\; x(s_P^{(g)}\Delta P + s_T^{(g)}\Delta T) + (1-x)(s_P^{(r)}\Delta P + s_T^{(r)}\Delta T) + \Delta x(Q_0^{(g)} - Q_0^{(r)}). \tag{8}$$

In this linear-response approximation, the terms that are higher order in $\Delta$ are ignored. The implicit assumptions in this
formulation are: 1) the climate sensitivities of runoff from glacierised and non-glacierised parts do not change appreciably over
a few decades, and 2) the contribution of a changing $x$ to changes in summer runoff is well represented by the recent difference
between the mean runoff of the glacierised and non-glacierised parts. We note that a similar linear-response approache been
used to analyse glacier-compensation effect (Chen and Ohmura., 1990), but without explicitly referring to climate sensitivity.

For Periche sub-catchment of upper Dudhkoshi catchment, possible changes in total runoff by 2050 were studied previously
using hydrological model simulation (Soncini et al., 2016). We used the corresponding projected changes in $P$, $T$ and $x$ to
obtain the summer runoff changes using Eq. (8) described above. These climate-sensitivity based predictions were compared
with the available estimates from hydrological model simulations (Soncini et al., 2016) assuming the recent ratio of winter to
summer runoff to remain unchanged.

We investigated the ability of Eq. (8) to predict the timing and the magnitude of the 'peak water' (Hock et al., 2005; Huss
and Hock, 2018) in the studied catchments. For these computations, the projected future changes in glacier area for Ganga
and Indus basin for three different climate scenarios (RCP 2.6, 4.5, and 8.5) (Huss and Hock, 2018) were applied to Chandra,





and upper Dudhkoshi catchments, respectively. The corresponding future temperature changes were obtained from projections available for the western and eastern Himalaya, respectively (Kraaijenbrink et al., 2017). The precipitation changes were not significant within the uncertainties (Kraaijenbrink et al., 2017), and were ignored here. The above estimates of the magnitude and the timing of the peak water were compared with the corresponding gridded values available in the literature (Huss and Hock, 2018). For this comparison of estimated peak-water, we considered the future changes in runoff of the parts of the

catchments that were glacierised at 2000, following the convention of Huss and Hock (2018).

### 3.2 Hydro-meteorological and glaciological input data

#### 3.2.1 Observations

Observed hourly runoff of Chandra river at Tandi (32.55°N, 76.97°E, 2850 m a.s.l.) from 26th June, 2016 to 30th Oct, 2018 ((Singh et al., 2020); see supplementary Table S1 for details) was available for three summer seasons with some data gaps

(Fig. 3b). Hourly 2m air temperature, precipitation, and incoming shortwave radiation were measured at the Himansh station (32.409°N, 77.609°E, 4080 m a.s.l.) in the catchment between 18th October, 2015 to 5th October, 2018 with some data gaps (supplementary Table S1).

Hourly runoff from upper Dudhkoshi catchment was observed at Phadking (27.74°N, 86.71°E, 2600 m a.s.l.) between 7th April, 2010 and 16th April, 2017 (Fig. 3a) (Chevallier et al., 2017). Available hourly air temperature and precipitation data

at Phadking from 7th April, 2010 to 23rd April, 2017 (with some data gaps) (Chevallier et al., 2017) were used. The daily incoming shortwave radiation data for the period 1st November, 2010 to 30th November, 2014 at nearby Changri Nup station (27.983° N, 86.783° E, 5400 m a.s.l.) in the same catchment were used ((Sherpa et al., 2017) ; see supplementary Table S1 for details).

Randolph Glacier Inventory (RGI 6.0) (Arendt et al., 2017) was used for the glacier boundaries in both the catchments. The

boundaries corresponded to the glacier extent in 2002. We considered 8 available geodetic mass-balance observations for each of the catchments that spanned a decade or more (see supplementary Table S3 for details).

#### 3.2.2 Reanalysis data and bias correction

We used hourly 2m air temperature, precipitation, and wind-speed from fifth-generation European Center for Medium-Range Weather Forecasts Atmospheric Reanalysis of the global climate (ERA5) from 1980 to 2018 (Hersbach et al., 2020), at a spatial

resolution of 0.25°×0.25°, as model inputs. Following the existing hydrological studies of various high Himalayan catchments (Immerzeel et al., 2013; Khadka et al., 2014; Soncini et al., 2016; Shrestha et al., 2017; Azam and Srivastava, 2020), the precipitation and temperature data were bias corrected as discussed below.

The available observed air temperature data at the Himansh station (Chandra catchment), and at Phadking (Dudhkoshi Catchment) were used to compute the mean monthly temperature biases for both the catchments (supplementary Fig. S1). The

computed biases were corrected for, assuming they remain the same for the whole catchment and over the whole simulation





period. While comparing mean monthly temperature of the stations and that of the corresponding grids of ERA5, mean monthly lapse rates (supplementary Fig. S2) were used to correct for the station elevation.

To correct for possible biases in the ERA5 precipitation data, a constant scale factor $\alpha_P$ was used for each of the catchments following the existing studies from the region (e.g. Bhattacharya et al., 2019; Krakauer et al., 2019; Zaz et al., 2019; Kanda et al., 2020; Azam and Srivastava, 2020). The scale factor was calibrated using the observed runoff and glacier mass-balance data employing a Bayesian procedure as described in Sect. 3.2.4. In some of the existing studies from the region, an elevation-dependent precipitation scaling have also been employed (e.g. Immerzeel et al., 2013). As an elevation-dependent correction may potentially introduce additional uncertainties (e.g. Johnson and Rupper, 2020), we preferred to use a constant $\alpha_p$ for each catchments, keeping the number of calibration parameters to a minimum.

We scaled the incoming shortwave radiation obtained from VIC model by a catchment specific constant to match the corresponding mean values observed at Himansh (Chandra catchment) and Changri Nup (upper Dudhkoshi catchment) stations (supplementary Fig. S3).

### 3.2.3   Details of the hydrological model

We used VIC model (version 4.2.d, accessible from https://vic.readthedocs.io/en/master/; Liang et al. (1996)), forced by bias-corrected ERA5 precipitation and temperature reanalysis data, to solve the energy and water balance and simulate runoff for the above two catchments. Within this model, the partitioning of rain and snow was done based on a threshold temperature of $0°$ C (supplementary Table S2).

The area-elevation distributions for both the catchments were obtained from 30 m resolution Shuttle Radar Topography Mission digital elevation model (Farr et al., 2007). We used 10 elevation bands for each grid box. Depending on the elevation range within a grid box, the size of the bands varied in the range 100–300 m. Soil data were from Harmonized World Soil Database (Nachtergaele et al., 2010), land-use data were from Moderate Resolution Imaging Spectroradiometer (Friedl and Sulla, 2019), and vegetation data were from Land Data Assimilation System (Rodell et al., 2004). We ran the VIC model at a spatial resolution of $0.25° \times 0.25°$, and time resolution of an hour. The model resolution was dictated by that of ERA5 input data. Chandra and upper Dudhkoshi catchments covered parts of 11 and 6 ERA5 grid boxes (Figs. 1a–1b), respectively. The fractional grid cover varied from 2.5–92% (2–68%) for Chandra (upper Dudhkoshi) catchment. VIC model has an in-built feature to take care of fractional coverage of grid boxes.

While the two studied catchments have more than 20% glacier cover, VIC model does not have the capability to compute glacier melt or the evolution of the glacier extent (Liang et al., 1996). Here we augmented VIC model with a temperature-index glacier mass-balance module (Hock, 2003). As the observed shrinkage of glacierised fraction was small (0.01–0.05) over the simulation period for both the catchments (Table 1), we assumed a static glacier cover. Biases due to such static-glacier approximation was shown to be small in another glacierised Himalayan catchment over the same period (Azam and Srivastava, 2020). Note that a dynamic description of glaciers would, however, be essential for predicting the long-term future changes in runoff as large changes in glacier extent are expected over the coming decades (e.g. Kraaijenbrink et al., 2017).



The hydrological fluxes over the non-glacierised parts were simulated with the standard VIC model. On the glacierised parts, a separate VIC model simulation was used to compute snow melt and snow-covered fraction in each elevation band. For the snow-free glacier surface in each elevation band, a temperature-index model (Hock, 2003) with a catchment-specific degree-day factor (DDF) obtained the ice melt. For melt calculations, temperature of the individual elevation bands were obtained from bias-corrected ERA5 described before using mean monthly lapse rates derived from the same data set (supplementary Fig. S2). The DDF values for each of the catchments were calibrated using a Bayesian procedure as described Sect.3.2.4. Glacier runoff was defined as the sum of snow melt, ice melt, and rainfall on the glaciers (e.g. Radić and Hock., 2014).

For model spin-up, we extended the meteorological input data back by repeating the data from 1980 to 1984. Subsequently, simulations were run over the period 1980–2018. Streamflow routing was implemented following Lohmann et al. (1998) to route the total surface runoff, baseflow, and meltwater fluxes generated from glacierised and non-glacierised parts of each grid box at an hourly time step.

Glacier mass balance was computed by subtracting the annual ice and snow melt over the glaciers from the corresponding total snowfall. Complex mass-balance processes like avalanches (Laha et al., 2017) and debris effects (Kraaijenbrink et al., 2017) were ignored for the sake of simplicity.

### 3.2.4 Model calibration

With limited observed runoff data, the calibration of a large number of tunable model parameters in complex hydrological models may lead to issues like equifinality (Beven and Freer., 2001; Beven., 2006) and over-fitting, compromising the ability of the model to capture the corresponding climate response accurately. Therefore we avoided calibrating a large set of model parameters (Jost et al., 2012) in the present study. We calibrated for only two model parameters: 1) catchment-wide precipitation scale factor $\alpha_P$, and 2) DDF of the glacier melt model. These two key parameters determine the catchment wide water balance, and the glacier mass balance. For the rest of the VIC model parameters, we used the central values of the recommended range (supplementary Table S2). Note that calibrated VIC model parameters used in the existing studies from the region (Zhao et al., 2015, 2019) were comparable to the ones used here to within 10–20%, suggesting that the VIC model parameters used in the present study were realistic.

To calibrate for the parameters $\alpha_P$ and DDF, we used the following Bayesian approach (e.g. Tarantola, 2005). For given a set of available observations $d$ and a set of model parameters $\theta$, the posterior probability of the model parameters given the observations was,

$$p(\theta|d) \propto p(d|\theta)p(\theta). \tag{9}$$

Here $p(\theta)$ was the prior distribution of the model parameters $\alpha_P$ and DDF. A uniform prior distribution over a range covering the corresponding values reported over the High Mountain Asia: 0.6–2.6 for $\alpha_P$ (Krakauer et al., 2019; Zaz et al., 2019; Azam and Srivastava, 2020; Kanda et al., 2020), and 2–16 mm °C$^{-1}$ day$^{-1}$ for DDF (Singh et al., 2000; Zhang et al., 2006; Azam et al., 2019; Kayastha et al., 2020). The conditional probability of the observations $d$ given the model parameter $\theta$, $p(d|\theta)$, was





assumed to be,

$$p(d|\theta) \sim e^{-\frac{\sum_i (Q_i^{mod} - Q_i^{obs})^2}{2\sigma_Q}} \times e^{-\frac{1}{2}\frac{\sum_j (b_j^{mod} - b_j^{obs})^2}{2\sigma_b}}. \tag{10}$$

Here the superscript $obs$ ($mod$) denoted the observed (modelled) values for a given $\theta$. $Q_i$ was the total summer runoff for the $i$-th year, and the summation was over all the years with observed runoff data for a given catchment. The uncertainty $\sigma_Q$ in

summer runoff was taken to be 10% of the mean summer runoff based on existing estimates for other Himalayan rivers which varied from 5% to 10% (Haritashya at al., 2010; Srivastava et al., 2014; Kumar et al., 2016). $b_j$ was the observed regional geodetic glacier mass balance, with the index $j$ denoting individual records. The corresponding uncertainties were in the range 0.05–0.32 m w.e yr$^{-1}$ (supplementary Table S3). The empirical factor of $\frac{1}{2}$ in the exponent associated with the mass-balance observations in Eq. (10) ensured that the two exponential weights had similar magnitude for the most-probable model (e.g.

Tarasova et al., 2016; Meyer et al., 2019).

For each of the catchments, a total $11 \times 29$ (319) model runs scanned the two-dimensional parameter space with step sizes of 0.2 for $\alpha_P$, and 0.5 mm °C$^{-1}$ day$^{-1}$ for DDF. These runs mapped out $p(\theta|d)$ over the parameter space. The most-probable pair of model parameters, i.e., the values of $\alpha_p$ and DDF for which $p(d|\theta)$ was the maximum, was used for simulating the runoff and glacier mass balance. The weighted ensemble of all the 319 models obtained the corresponding $2\sigma$ uncertainties.

Due to the limited observed runoff data, the full runoff data set were utilised for the above calibration procedure without any validation period. As an additional check, for upper Dudhkoshi catchment, the calibration was performed using data from four consecutive years, with remaining three year's data utilised for validation. This experiment was repeated 4 times using runoff data from different sets of four consecutive years for calibration.

Following earlier studies (e.g. He and Pang., 2015; Isenstein et al., 2015), the parameter sensitivity of simulated results was

estimated with the help of additional 22 simulations where one of the 11 VIC model parameters (supplementary Table S2) was perturbed by ±25% of the range of corresponding recommended values. Perturbing the parameters one by one to capture the local sensitivity in the 11-d parameter space is similar to computing the multidimensional gradient in this space. To confirm the validity of this approach where only one of the model parameters was perturbed, additional 80 simulations were ran with a randomly chosen pair of parameters perturbed simultaneously.

## 4 Results and discussions

### 4.1 The calibrated models

The Bayesian calibration procedure obtained best-fit DDF values of 5.0 and 7.5 mm day$^{-1}$ °C$^{-1}$ for Chandra and upper Dudhkoshi catchments, respectively. These best-fit DDF values were in the same ballpark range as previously used in studies in and around Chandra (Azam et al., 2014, 2019; Pratap et al., 2019) and Dudhkoshi catchments (Pokhrel et al., 2014; Khadka

et al., 2014; Nepal, 2016). The best-fit $\alpha_P$ was 1.4 for both the catchments which was within the range of values 0.8–2.2 used in the existing studies in the Himalaya to correct various reanalysis products (Bhattacharya et al., 2019; Krakauer et al., 2019; Zaz et al., 2019; Azam and Srivastava, 2020; Kanda et al., 2020).



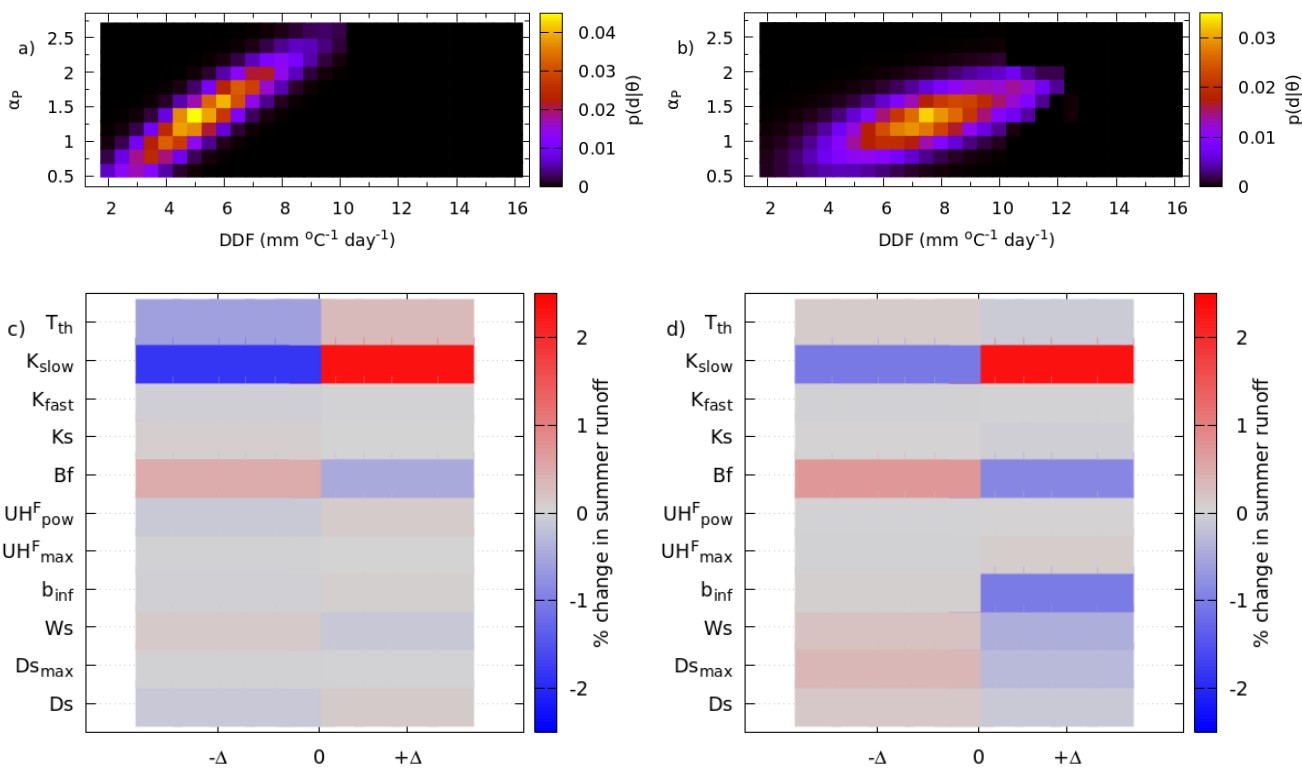

**Figure 2.** (a) and (b) shows the posterior probability distribution $p(d|\theta)$ of the model parameters ($\alpha_P$, DDF) for Chandra and upper Dushkoshi catchment, respectively (see Sect.3.2.4). (c) and (d) shows the sensitivities of the simulated summer runoff to perturbations in 11 VIC the model parameters for Chandra and upper Dushkoshi catchments, respectively. Here, $\pm\Delta$ denotes the perturbation of parameters by $\pm 25\%$ of the corresponding prescribed range (see Sect. 3.2.4 and supplementary Table S2).



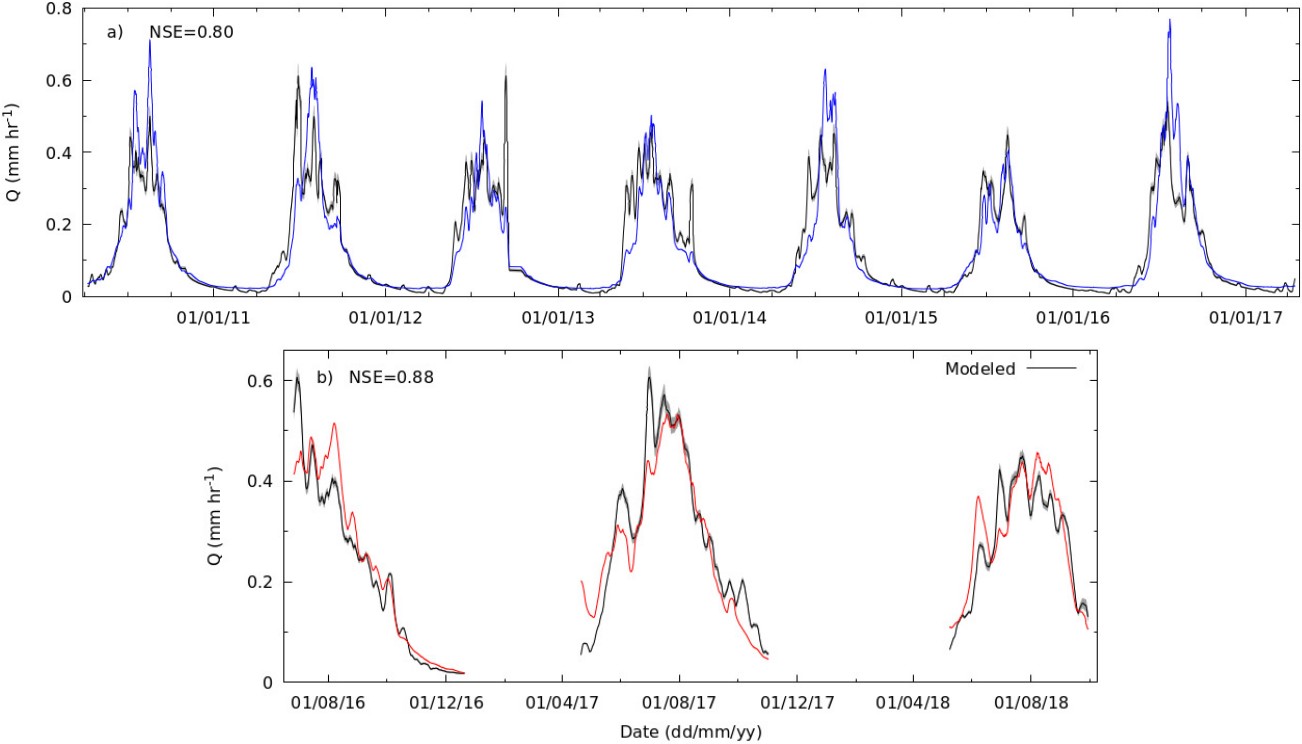

**Figure 3.** Modelled weekly runoff (black lines, with grey bands denoting 2-$\sigma$ uncertainty) compared with the corresponding observations for (a) upper Dudhkoshi (blue solid line), and (b) Chandra (red solid line) catchments.

The Bayesian calibration procedure which simultaneously fitted the glacier mass balance and the summer runoff data, yielded unique best-fit models for the catchments studied (Figs. 2a–2b). In contrast, the existing studies from the region where DDF was calibrated using only discharge data, usually encountered equifinality issues (e.g. Azam and Srivastava, 2020) such that the best-fit model was not uniquely determined.

The calibrated models reproduced the observed summer runoff reasonably well (Fig. 3) with RMSEs of 11–12% of the mean summer runoff, and NSEs of 0.88–0.80 for the two catchments. The RMSEs and NSEs obtained here are comparable or smaller than those reported in the existing studies from the region (Nepal, 2016; Mimeau et al., 2018; Bhattacharya et al., 2019; Azam et al., 2019; Azam and Srivastava, 2020). Four additional calibration experiments for upper Dudhkoshi catchment, each one using a different set of 4 consecutive years of data for calibration, obtained comparable best-fit values of DDF (7.2±1.1 mm day$^{-1}$ °C$^{-1}$) and $\alpha_P$ (1.43 ± 0.03), with NSEs and RMSEs similar to those mentioned above.

### 4.2 Simulated runoff and its parameter sensitivity

The simulated mean summer runoff of Chandra and upper Dudhkoshi catchments were 1.08±0.05 m yr$^{-1}$ and 0.81±0.04 m yr$^{-1}$, respectively (supplementary Fig. S5). The mean summer runoff of the glacierised and the non-glacierised parts of





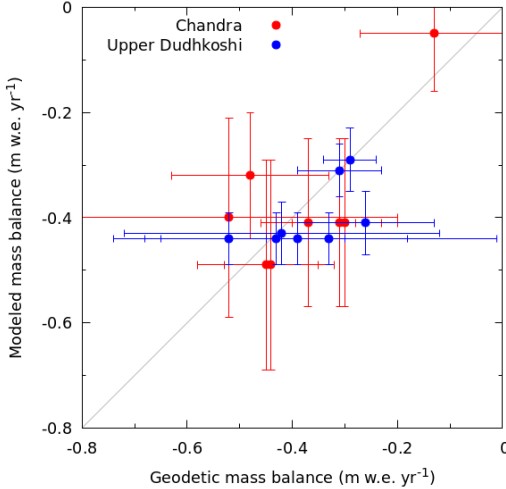

**Figure 4.** A comparison of the modelled glacier mass balance with the available regional geodetic mass balance for Chandra (solid red circles) and upper Dudhkoshi (solid blue circles) catchments. The solid gray line is the 1:1 reference line. See supplementary Table S3 for further details.

Chandra catchment were 1.54 and 0.92 m yr$^{-1}$, respectively. The corresponding values for upper Dudhkoshi catchment were 1.59 and 0.61 m yr$^{-1}$. In these two catchments more than 82% of the simulated annual runoff were during the summer season. In comparison, seven years of observation from upper Dudhkoshi catchment (Chevallier et al., 2017) showed a mean specific summer runoff of 0.86±0.05 m yr$^{-1}$ which was 83% of the mean annual runoff. The simulated mean annual runoff of upper

Dudhkoshi catchment was comparable to those reported in the existing studies from the region. For example, the estimated mean annual runoff of the Periche sub-catchment of upper Dudhkoshi was 0.95 m yr$^{-1}$ during 2013–2015 (Mimeau et al., 2018). The available estimate of the mean annual runoff of Dushkoshi catchment up to 670 m a.s.l. was somewhat higher (1.6 m yr$^{-1}$) due to a higher rainfall in the lower reaches (Nepal, 2016). No previous runoff estimate was available of Chandra catchment.

In the sensitivity analysis, the absolute changes in summer runoff were less than $\sim 1.5\%$ for the perturbed values of all but two parameters (Figs. 2b–2d). The slightly higher summer-runoff sensitivities of 1.8–2.5 % for the longer time scales in the routing model became less than 1% When the annual runoff was considered. In the additional 80 simulations where two parameters were perturbed simultaneously, the resultant runoff sensitivities were approximately equal to the sum of those obtained in the corresponding pair of simulations with a single perturbed parameter (supplementary Fig. S4).

**4.3   Simulated glacier mass balance and its climate sensitivity**

The simulated glacier mass balance for Chandra (upper Dudhkoshi) catchment over 1980–2018 was $-0.18\pm0.10$ m w.e. yr$^{-1}$ ($-0.37\pm0.04$ m w.e. yr$^{-1}$), which was comparable to the existing geodetic observations within the uncertainties (Fig. 4;



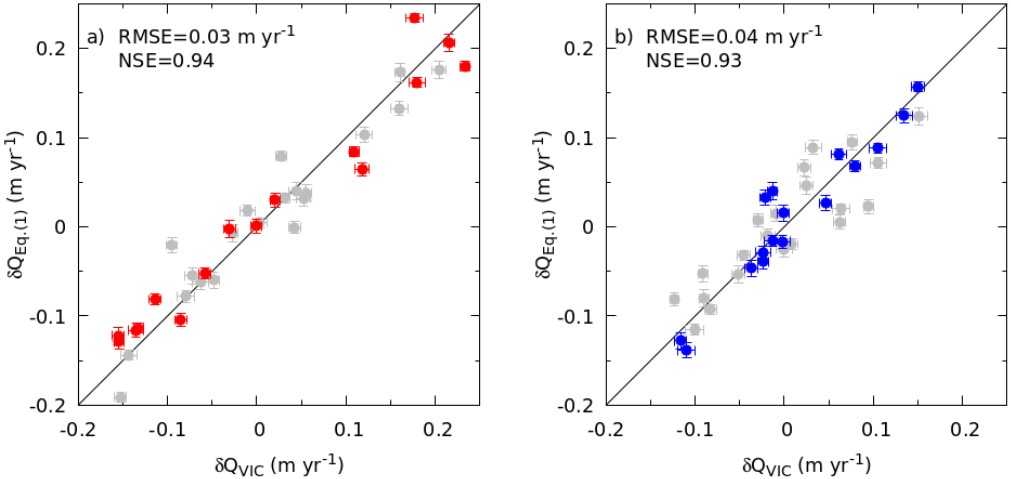

**Figure 5.** The summer runoff anomalies computed using the Eq. (1) ($\delta Q_{\text{Eq.}(1)}$) and those from the VIC model simulations ($\delta Q_{\text{VIC}}$) are compared for (a) Chandra, and (b) upper Dudhkoshi catchments. The gray solid circles denote data from the calibration period 1997–2018. The red (blue) solid circles denote data from Chandra (upper Dudhkoshi) catchment over the validation period 1980–1996. The solid diagonal line denotes a perfect match for reference.

supplementary Table S3). The RMSE between modelled and observed mass balance of Chandra (upper Dudhkoshi) catchment was 0.10 m w.e. yr$^{-1}$ (0.11 m w.e. yr$^{-1}$). Our simulations indicated that glacier runoff contributed 39±9% and 36±11% of the total summer runoff in upper Dudhkoshi and Chandra catchments, with th glacier ice loss amounting to 9% and 4% of the respective total summer runoff. The estimated glacier contributions to runoff were largely consistent with the existing studies from the region when differences in fractional glacier cover were taken into account (supplementary Table S4).

The sensitivity of the modelled glacier mass balance to temperature was $-475 \pm 93$ and $-274 \pm 46$ mm yr$^{-1}$ °C$^{-1}$ for Chandra and upper Dudhkoshi catchments, respectively. The corresponding precipitation sensitivities for these catchments were $200 \pm 42$ and $49 \pm 20$ mm yr$^{-1}$ for a 10% change in precipitation. These sensitivities were all significant at $p < 0.01$ level, and were consistent with the known estimates for Chhota Shigri (Azam et al., 2014) and Zhadang (Mölg et al., 2012) glaciers, and available regional-scale values (Shea and Immerzeel, 2016; Wang et al., 2019) (supplementary Table S5). Only the reported value for Dokriani Glacier from the central Himalaya (Azam and Srivastava, 2020) was exceptionally high in comparison.

### 4.4 Variability of summer runoff and linear response

During 1980–2018, the simulated summer runoff in Chandra and upper Dudhkoshi catchments varied in the range 0.86–1.33 and 0.55–0.98 m yr$^{-1}$, respectively. The corresponding standard deviation of summer runoff were 0.14, and 0.10 m yr$^{-1}$ for Chandra and upper Dudhkoshi catchments, respectively. The respective ranges of summer temperature were 2.0–5.3 and 1.2–





2.3°C, and those of annual precipitation were 1.05–2.10 and 1.17–1.92 m yr$^{-1}$. A linear fit of the summer runoff anomalies

to those of summer temperature and annual precipitation during 1997–2018 worked well for Chandra ($R^2$=0.92) and upper Dudhkoshi ($R^2$=0.93) catchments. The best-fit form for the calibration period 1997–2018 reproduced the variability of summer runoff over the validation period 1980–1996 reasonably well (Fig. 5) for both the catchments with RMSE < 0.04 m yr$^{-1}$ and NSE > 0.93. These results indicated that a linear-response framework (Eq. (1)) provided a good description of the interannual variability of summer runoff in these catchments. We also confirmed that the interannual variabilities of annual precipitation

and summer temperature were uncorrelated in these catchments at $p < 0.05$ level.

### 4.5 Climate sensitivities of catchment runoff

The best-fit temperature sensitivities of summer runoff $s_T$ were 117±8 and 116±34 mm yr$^{-1}$ °C$^{-1}$ for Chandra and upper Dudhkoshi catchments, respectively. The best-fit $s_P$ were 0.39±0.03 and 0.47±0.06 mm yr$^{-1}$ mm$^{-1}$ for Chandra and upper Dudhkoshi catchments, respectively. These sensitivities were all significant at $p < 0.01$ level. Note that $s_T$ and $s_P$ for both the

catchments were the same up to the uncertainty, despite the differences in their climate setting. Also, in both the catchment $s_P$ was significantly smaller than 1 mm yr$^{-1}$ mm$^{-1}$, possibly indicating that an interannual change of the snow/ice storage in response to precipitation changes dampened of the precipitation sensitivity of summer runoff.

To the best of our knowledge, only two other studies from the Himalaya attempted to quantify the temperature sensitivity of catchment runoff. In Dokriani glacier, the reported $s_T$ of 620 mm yr$^{-1}$ °C$^{-1}$ (Azam and Srivastava, 2020) was signifi-

cantly higher than the estimates presented above. This was possibly due to the exceptionally high temperature sensitivity of glacier mass balance estimated for the glacier as discussed before, and a relatively higher glacierised fraction of 0.5. Previously reported $s_T$ from Dudhkoshi catchment was $5.7 \pm 0.3$ mm yr$^{-1}$ °C$^{-1}$ (Pokhrel et al., 2014) - an order of magnitude smaller compared to that obtained here and by Azam and Srivastava (2020). Our estimates of $s_P$ from both Chandra and upper Dudhkoshi catchments were comparable with those reported earlier (Pokhrel et al. (2014); Azam and Srivastava (2020);

supplementary Table S6).

### 4.6 Climate sensitivities of glacier runoff

Our analysis of runoff from the glacierised part of the catchments obtained $s_T^{(g)}$ of 405±24 and 469±61 mm yr$^{-1}$ °C$^{-1}$ for Chandra and upper Dudhkoshi catchments, respectively, which were significant at $p < 0.01$ level. The corresponding $s_P^{(g)}$ values were $-0.12\pm0.08$ and $0.00\pm0.02$ yr$^{-1}$ mm$^{-1}$, none of which were significant at $p < 0.05$ level. Note that both the

catchments had similar values of $s_P^{(g)}$ and $s_T^{(g)}$ within the limits of uncertainty despite their contrasting climate setting.

Interestingly, summer runoff from both winter-accumulation type glaciers in Chandra catchment and summer-accumulation type glaciers in upper Dudhkoshi catchment was approximately independent of the corresponding precipitation variability. This precipitation-independent glacier runoff is a novel finding that may not has been discussed in the existing glacio-hydrological studies in the Himalaya. Since the patterns of the climate sensitivities of the glacier runoff discussed above were similar for the

two catchments despite their contrasting climate regime, it may be worth exploring if this is a general feature of the climate sensitivity of glacier runoff in the Himalaya or elsewhere.





According to our simulations, snowfall constituted 97% and 86% of the total precipitation over the glaciersied parts of Chandra and upper Dudhkoshi catchments, respectively. Consequently, a positive precipitation anomaly did not translate into a higher summer runoff of the glaciers. For the years with high precipitation (equivalently, high snowfall), snow and ice melt output from the glaciers reduced due to a higher albedo, which reduced the available melt energy. This led to a negative correlation ($r = -0.5$, $p < 0.05$) between simulated summer runoff and precipitation in these two catchments. These effects made the runoff of the glaciersied parts relatively insensitive to precipitation variability (upper Dudhkoshi catchment) or have a small negative precipitation sensitivity (Chandra catchment). The latter was likely related to a higher snow-to-rain ratio in the winter snow dominated Chandra catchment where any change in precipitation leads to a opposite change in glacier melt. For example, in years with a higher snowfall the glacier melt was suppressed due to a strong albedo feedback, leading to a slight reduction of the glacier runoff.

The negligible $s_P^{(g)}$ discussed above implied a stabilisation of the total runoff of the glacierised catchments against precipitation variability, as the runoff contribution from the glacierised fraction $x$ was essentially independent of precipitation. The magnitude of this stabilising effect scaled with the glacier fraction ($x$) in the catchment. The above stabilising effect is consistent with the reported buffering of catchment runoff by glaciers during the extreme drought years across High Mountain Asia (Pritchard, 2019).

The relatively high $s_T^{(g)}$ in both the catchments as presented above can be understood as follows. A higher temperature implied a higher available energy leading to higher meltwater flux from the glaciers. The glaciers effectively acted as infinite reservoirs over an annual scale so that the meltwater volume was limited only by the available energy. These arguments were consistent with a high correlation ($r > 0.9, p < 0.05$) between the summer temperature and summer runoff of the glacierised parts for both the catchments.

### 4.7 Climate sensitivity of runoff from the non-glacierised parts

In the non-glacierised parts of Chandra and upper Dudhkoshi catchments, $s_T^{(r)}$ of 21±13 and 28±39 mm yr$^{-1}$ °C$^{-1}$, and $s_P^{(r)}$ of 0.56±0.04 and 0.59±0.07 mm yr$^{-1}$ mm$^{-1}$ were obtained. These sensitivities were all significant at $p < 0.01$ level.

Again, both the catchments had similar values of $s_P^{(r)}$ and $s_T^{(r)}$ within the corresponding uncertainties. However, compared to the glacierised parts, the climate sensitivities of the runoff from the non-glacierised parts showed an exactly opposite trend with summer runoff relatively insensitive to temperature anomalies and sensitive to precipitation anomalies. Because of the presence seasonal snow cover over the non-glacierised parts, a temperature dependence of the summer runoff may be expected. However, the amount of snowmelt was limited by the supply of seasonal snow, and not by the available energy. This led to a weak response of the total summer runoff from the non-glacierised parts to temperature forcing. This argument was supported by the fact that summer runoff from the non-glacierised parts were uncorrelated with summer temperature and strongly correlated with summer precipitation ($r > 0.9, p < 0.05$).





### 4.8 Implications for future changes in the mean and variability of summer runoff

The estimated climate sensitivities from glacierised and non-glacierised parts of both the catchments suggested $s_P^{(g)} \approx 0$ and
$s_T^{(r)} \approx 0$. With these approximations, eqs. 1–8 simplified to the following approximate relations describing the response of the
summer runoff to climate variability and change.

$$
\delta Q \approx x s_T^{(g)} \delta T + (1-x) s_P^{(r)} \delta P, \tag{11}
$$

$$
\delta Q^{(g)} \approx x s_T^{(g)} \delta T, \tag{12}
$$

$$
\delta Q^{(r)} \approx (1-x) s_P^{(r)} \delta P, \tag{13}
$$

$$
\sigma_Q \approx \sqrt{x^2 s_T^{(g)2} \sigma_T^2 + (1-x)^2 s_P^{(r)2} \sigma_P^2}, \tag{14}
$$

$$
\Delta Q \approx x s_T^{(g)} \Delta T + (1-x) s_P^{(r)} \Delta P + \Delta x (Q_0^{(g)} - Q_0^{(r)}). \tag{15}
$$

These simplified equation suggested that the key parameters that determine the climate reponse of the catchment to gievn
forcing were $s_T^{(g)}$ and $s_P^{(r)}$ . Below we first analyse the predictive power of Eq. (15), and then discuss the implications of the
above simplified linear-response description in the context glacier-compensation effect (e.g. Chen and Ohmura., 1990) and
peak-water effect (e.g. Huss and Hock, 2018) for the catchments.

#### 4.8.1 The future changes in mean summer runoff

We tested the accuracy of $\Delta Q$ estimated using Eq. (15) for Periche sub-catchment of upper Dudhkoshi catchment. Previously
reported projections for Periche catchment suggested changes in annual precipitation, annual temperature, and glacier cover of
A) 0.01 m, 1.25°C, and 0.12, and B) 0.02 m, 1.45°C, and 0.12, respectively, by 2050 (Soncini et al., 2016). The corresponding
reported changes in summer runoff were A) 0.10, and B) 0.12 m yr$^{-1}$ (Soncini et al., 2016) . For the above to scenarios,
Eq. (15) yielded similar summer runoff changes of A) 0.08±0.01, and B) 0.10±0.02 m yr$^{-1}$. This indicated the reliability of
the climate sensitivities of summer runoff presented here, and also confirmed that accurate climate sensitivities can be used for
an accurate prediction of catchment response to given forcing over multidecadal scales with minimal computational cost (e.g.,
Vano and Lettenmaier, 2014).

#### 4.8.2 Future changes in the variability of summer runoff and glacier-compensation effect

Consider a set of hypothetical catchments with different values of $x$, but similar values of $s_T^{(g)}$, $s_P^{(r)}$, $\sigma_T$ and $\sigma_P$. Then, Eq. (14)
implies that $\sigma_Q$ is a non-monotonic function of $x$ (Fig. 6a). It has a high value in the limit $x \to 0$ due to a strong precipitation
sensitivity of the off-glacier runoff, with $\sigma_Q \approx (1-x) s_P^{(r)} \sigma_P$. In the opposite limit of $x \to 1$, $\sigma_Q$ is again high due to a high
temperature sensitivity of glacier runoff, with $\sigma_Q \approx s_T^{(g)} \sigma_T$. These to competing effects imply a minimum value of $\sigma_Q$ at
an intermediate value of $x$ (Fig. 6a). This is the well known glacier compensation effect (e.g., Chen and Ohmura, 1990).
Thus, Eq. (14) provides a theoretical explanation of the glacier-compensation effect. Note that Eq. (14) suggests a hyperbolic
behaviour of $\sigma_Q(x)$, while some of the existing studies used an empirical parabolic curve to describe the depndence of the
coefficient of variation of runoff from glacierised catchments on $x$ (e.g., Chen and Ohmura., 1990). The results discussed





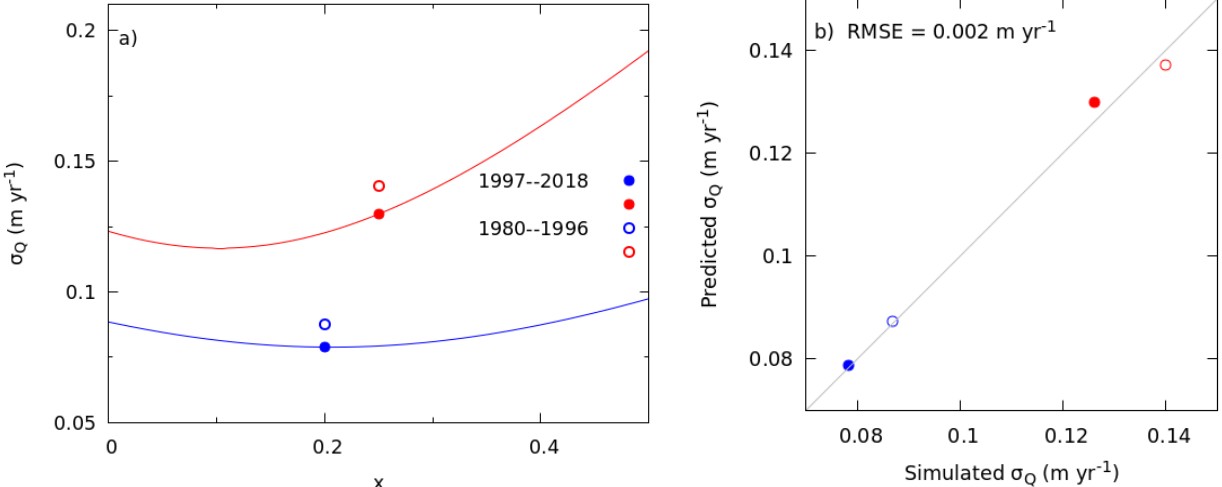

**Figure 6.** a) The solid and open circles denote the simulated summer runoff variability $\sigma_Q$ for the periods 1997–2018, and 1980–1996, respectively, for Chandra (red symbols) and upper Dudhkoshi (blue symbols) catchments. The solid lines indicate the expected behavior of $\sigma_Q$ as function of glacier fraction $x$, if $\sigma_T$ and $\sigma_T$ are assumed to remain fixed at their corresponding values during 1997–2018. b) The simulated $\sigma_Q$ is compared with the corresponding predictions using Eq. (14) (see text for details).

above are also consistent with a reported strong correlation between runoff and precipitation (temperature) in the limit of small
(extensive) glacier cover (Van Tiel et al., 2020).

It was proposed by Chen and Ohmura. (1990) that the parabolic dependence of the coefficient of variation of runoff on glacierised fraction can be utilised to estimate the change in $\sigma_Q$ for any given change in glacier cover. However, it has been reported that the corresponding best-fit parabola changes with time, making such predictions unreliable (Van Tiel et al., 2020). This effect can be explained based on Eq. (14) as follows. It is apparent from Eq. (14) that together with the glacier cover $x$, if
$\sigma_P$ and $\sigma_T$ were to change under a changing climate, then the Eq. (14) describes a different hyperbolic curve altogether. This can be verified from fig. 6a: the simulated $\sigma_Q$ for both the catchments for the period 1980-1996 were higher than those during the period 1997-2018, even as $x$ remained constant in our simulations. These changes were due to differences in $\sigma_P$ and $\sigma_T$ in the corresponding period. Note that the simulated $\sigma_Q$ values during 1980–1996 could be predicted well using Eq. (14) together with $\sigma_P$ and $\sigma_T$ values over the same period (Fig. 6b). In fact, it is apparent from Fig. 6a that for the two studied catchments,
the future changes in $\sigma_Q$ are likely to be dictated by those of climate variability (i.e., $\sigma_P$ and $\sigma_T$). The influence of a changing glacier cover on the runoff variability may be relatively minor. Overall, the above exercise indicated that Eq. (14) may be used to quantify the changes in the variability of future runoff, and the departures of observed/modelled catchment response from the glacier-compensation-effect curve (Van Tiel et al., 2020).





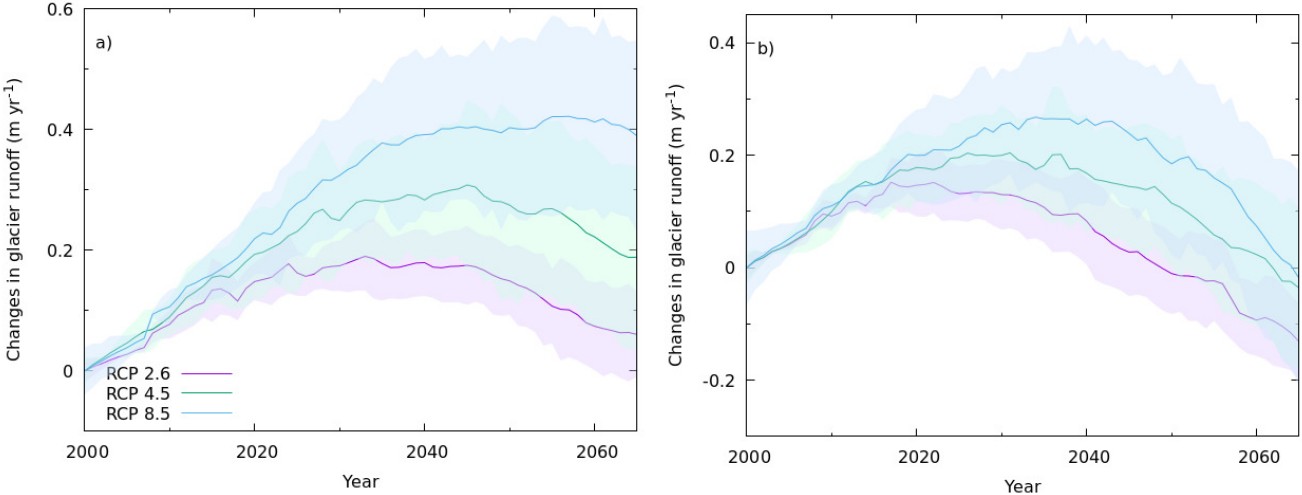

**Figure 7.** The 'peak water' due to future glacier changes predicted for three different climatic scenarios using Eq. (15) for (a) Chandra, and (b) upper Dudhkoshi catchments, respectively. See text for details.

### 4.8.3 Peak water due to shrinking glaciers

The future changes of glacier runoff from Chandra and upper Dudhkoshi catchments as obtained from Eq. (15) captured the 'peak water' successfully (Fig. 7). For all the three climate scenarios (supplementary Fig. S6), an initial steady increase in the summer runoff from the glacier was followed by a steady decline. In Chandra catchment, the estimated timing of peak water were 2033, 2045, and 2055, respectively, for the RCP 2.6, 4.5, and 8.5 scenarios. The corresponding peak glacier runoff were 1.73, 1.85, and 1.97 m yr$^{-1}$, compared to the recent value of 1.54 m yr$^{-1}$ (supplementary Table S7). In upper

Dudhkoshi catchment, the estimated timing of peak water were 2022, 2031, and 2034, respectively, for the above scenarios. The corresponding peak summer runoff from the glaciers were 1.74, 1.80, and 1.86 m yr$^{-1}$, compared to the recent value of 1.59 m yr$^{-1}$ (supplementary Table S7). These predicted 'peak water' magnitude may not lead to a observable signal in the catchment runoff given that the present $\sigma_Q$ values of 0.08 (upper Dudhkoshi catchment) and 0.13 (Chandra catchment) m yr$^{-1}$ (Fig. 6a) in these catchments. Except for the more ambitious RCP 2.6 scenario, in other scenarios the 'peak water'

signal is likely to be observed in the catchment runoff over the background interannual variability of runoff ($\sigma_Q$) in these two catchments.

It is interesting that the simple climate-sensitivity based estimates obtained timing and amplitude of the 'peak water' that were comparable with the previously reported values (supplementary Table S7) computed using state-of-the-art glacio-hydrological simulations (Huss and Hock, 2018). In upper Dudhkoshi catchment, however Eq. (15) predicted somewhat quicker

and smaller 'peak water' compared to the previously reported value (supplementary Table S7). This inconsistency may be related to the use of different glacier inventories and meteorological forcing. For example, Huss and Hock (2018) obtained a specific glacier runoff in Ganga basin which was about twice as large as obtained here for upper Dudhkoshi catchment. Also,





Huss and Hock (2018) mentioned possible intensification of Indian summer monsoon over the eastern Himalaya, while precipitation changes were ignored in the present study due to the large uncertainty. We note that the present formulation (Eq. (15))

ignored the feedback of the evolving glacier geometry on mass balance, and thus, possible future long-term changes in $Q_0^{(g)}$. However, despite the limitations, the above exercise overall indicated that a climate-sensitivity based approach was able to capture the basic characteristic of the long-term response of glaciersied catchment, namely, the peak water effect, both qualitatively and quantitatively.

### 4.9   Model limitations

An obvious limitation of the present study is a lack of long-term hydro-meteorological field data. However, this is a common problem that affects glacio-hydrological studies in the Himalaya. We have used runoff time series generated with VIC model simulations to circumvent the issue. A bias correction of the reanalysis product using the available meteorological field data, the use of total summer runoff and the decadal-scale glacier mass balance data for calibration, a Bayesian calibration procedure, calibrating for only two parameters to avoid over-fitting ensured robustness of the model results presented. The model

sensitivity to the specific choice of parameters were generally low. Available estimates from the existing literature with the corresponding results presented here indicated reasonable model performance as well.

We reemphasise that the climate sensitivity discussed are only applicable over the range of forcing used in the corresponding fits. Over shorter time scale of multiple decades, as long as the changes in forcing is within the range of the fits, the climate sensitivities presented here may be expected to characterise the catchment response accurately. Also, over longer time scales

the dynamic glacier geometry which was ignored in our simulation may lead to systemtic changes in $Q_0^{(g)}$ that cannot be captured in a linear response approach.

## 5   Summary and conclusions

In this paper, we simulate the runoff of Chandra (western Himalaya) and upper Dudhkoshi (eastern Himalaya) catchments over 1980–2018, using VIC model augmented with a temperature-index glacier-melt module. With a minimal calibration strategy

of calibrating only two parameters to fit the available data of summer runoff and decadal-scale geodetic glacier mass balance, the model produces good match with both observations and the results available in the literature. The interannual variability of summer runoff simulated with the VIC model is then utilised to obtain the climate sensitivity of summer runoff to temperature and precipitation forcing in the catchments, which leads to the following conclusions.

- For both the catchments, summer runoff from the glacierised parts have a high temperature sensitivity, and that from the

non-glacierised parts is essentially temperature independent. Temperature sensitivity of summer runoff in Chandra and Upper Dudhkoshi catchments are $117 \pm 8$ and $116 \pm 34$ mm yr$^{-1}$ $^\circ$C$^{-1}$, respectively.



- Precipitation sensitivity of summer runoff from the non-glacierised parts are high, but that of the runoff from the glacierised parts is negligible. Precipitation sensitivity of summer runoff in Chandra and Upper Dudhkoshi catchments are $0.39 \pm 0.03$ and $0.47 \pm 0.06$ mm yr$^{-1}$ mm$^{-1}$, respectively.

- In the limit of low glacier cover, an increasing glacier cover stabilises catchment runoff against precipitation variability due to a precipitation-independent runoff of the glaciers. In contrast, at the limit of high glacier cover, the variability of catchment runoff increases with glacier cover due to a strong temperature depndence of glacier runoff. This leads to the so-called 'glacier-compensation effect'. The present climate sensitivities and the future variability of precipitation and summer temperature, will determine together the future variability of summer runoff.

- A knowledge of the climate sensitivities of runoff of the glacierised and the non-glacierised parts allows an estimation of future changes of catchment runoff and the amplitude and the timing of 'peak water', under any given climate scenario.

*Data availability.* All the observed hydro-meteorological data, basic simulation results, and codes developed will be made public.

*Author contributions.* AB conceived the study, and developed the theoretical framework. SL ran the simulations. SL and AB did the data analysis. AB wrote the manuscript with inputs from SL. AS, PS, and MT did the field measurements at Chandra catchment, and analysed the field data. All the authors participated in the discussions, and edited the manuscript.

*Competing interests.* The authors have no competing interests to declare.

*Acknowledgements.* The study is funded under the HiCOM initiative by ESSO-NCPOR, MoES (grant number NCAOR/2018/HiCOM/04). SL acknowledges INSPIRE Research fellowship (grant no IF 170526). AB acknowldges supprot from MoES grant no MoES/PAMC/H&C/79/2016-PC-II. We thank the French Agence Nationale de la Recherche (references ANR-09-CEP-0005-04/PAPRIKA and ANR-13-SENV-0005-03/PRESHINE) for sharing the hydro-meteorological data in the upper Dushkoshi catchment. We greatly benefited from discussions with Ela Chawla, Subimal Ghosh, Pradeep P Mujumdar, Raghu Murtugudde, and Vikram.



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
