# Peer review of "The control of climate sensitivity on variability and change of summer runoff from two glacierised Himalayan catchments"

_Hydrology and Earth System Sciences, 2021_

## Author Comment (AC1)

**Authors' response (AC1) to the comments by Reviewer #1 (RC1)**

Understanding the uncertainties of the runoff in the glacierised catchments and its impacts on the availability of water in Himalayan rivers is critical. This manuscript discussed the climate sensitive on variability and change of summer runoff in two glacierised Himalayan catchments. The authors argued that the runoff from the glacierised parts of the catchments is sensitive to temperature changes, but is insensitive to precipitation changes. With shrinking glacier cover over the coming decades, the summer runoff from the two catchments is expected become more sensitive to the precipitation forcing and less sensitive to the temperature forcing. These conclusion is clear and interesting, but not novel enough with comparison of the recent other researches. However, I still recommend this manuscript can be published.

We thank the reviewer for the critical comments. We decided to drop the adjective 'novel' in the revised version based on comments from both the reviewers.

Some suggests:

1. The description of the VIC model (including the input data, e.g. CMIP 5 data) is to simple toohelp for the reader to understand the article.
2. The authors did not considered the glacier volume change, then the conclusion is somewhat not convincible

1. We shall add additional details about the model and input data used. We shall add a flowchart to explain the model workflow better. The flowchart is added below. Please refer to AC2 for more details.

2. We did not consider the glacier changes over the period 1980-2018, as total glacier cover change was less than 5% of the catchment area. We have acknowledged and discussed this issue in L194–L195. We have referred the reader to the relevant reference that supports our assumption.

[Figure]

Fig 1. A schematic representation of the modelling strategy used in the present study.

---

## Author Comment (AC2)

**Authors' response (AC2) to the comments by Reviewer #2 (RC2)**

In this manuscript the climate sensitivities of runoff (catchment, glacierized and non-glacierized parts) in two contrasting glacierized Himalayan catchments are analyzed. The obtained climate sensitivities are then used to derive a measure for the standard deviation of streamflow and to make projections of how streamflow will change under climate change. A hydrological model combined with a glacier melt model is used to simulate streamflow timeseries for the two catchments. Instead of deriving the streamflow variability and the future changes directly from the model output, the model simulations are used to get the climate sensitivities of summer runoff to annual precipitation and summer temperature. This step is likely needed because the hydrological model only includes a static glacier. The study comes to the (not very novel) conclusion that glacierized parts of the catchment are sensitive to temperature and that the non-glacierized parts are sensitive to changes in precipitation. It also suggests that climatesensitivities can be used to estimate magnitude and timing of peak water,

but it is unclear how climate sensitivities should be derived for ungauged catchments and what the advantages are of not directly estimating peak water from (glacio)-hydrological models.

Overall, I think that, despite the interesting topic of climate sensitivity of glacierized catchments which is suitable for HESS, the study does not do a good job in addressing a clear research gap. Besides, I have some doubts about some parts of the methods, there are some unclear descriptions and there is a lack of discussion on the simplifications, interpretation of the findings and the implications. Please find below my major and minor Comments.

[1] Fundamentally climate sensitivity controls the response of any system to changes in climate, and allow an efficient quantification of the response. To give a few examples, climate sensitivity of glaciers was used 1) to invert for the century-scale global temperature history (Oerlemans, 2005), 2) to compute future sea-level changes (Leclerq et al., 2011), 3) to explain the spatial patterns of glacier loss in the Himalaya (Fujita & Sakai, 2017), etc. While state-of-the-art process-based models can directly predict the future, the climate sensitivities are useful in analysing such predictions. Underlying model assumptions or parameterisations may lead to biases in the projections, which can be identified using climate sensitivities. For example, an analysis of the modelled climate sensitivities helped identify and explain an inherent bias in scaling-based glacier evolution models commonly used for predicting the future changes in runoff and sea-level (Banerjee et al., 2020). That's why we believe a knowledge of the climate sensitivity of Himalayan runoff is necessary. While there are numerous studies of future changes in runoff in the glacierised Himalaya (Azam et al., 2021), a quantification and/or analysis of the climate sensitivities is largely missing. This is the research gap that the present paper addresses by computing and analysing the climate sensitivities of runoff from two Himayalan catchments. We apologise for not conveying the above points in the introduction clearly enough. We shall revise the introduction section accordingly.

We agree that there is nothing noble about a temperature-sensitive glacier runoff. However, a precipitation-independence of glacier runoff has not been reported/discussed previously in the Himalaya or elsewhere. However, keeping in mind both reviewers' view, we shall drop the adjective "novel". The other specific points raised above are clarified in our replies below.

Novelty and research gap
The introduction of the manuscript is very minimal, it touches on a few topics but does not show how this study fits in between previous studies and it does not clearly explain the research gap and what the study aims to achieve and why. The sentence 'Due to a lack of long-term data.... may still be lacking' does not do justice to all the studies that exist on streamflow and its projections of Himalayan catchments.
Here I would expect to read what climate sensitivities can add to the existing (modelling) studies.
Then in the second part of the introduction, the explanation of how climate sensitivities are related to long-term changes and the glacier compensation effect are very unclear. What to do with the sentence 'Climate-sensitivity based predictions for future changes in runoff are reliable ..........over the calibration period'? How does that match with the peak water exercise in the manuscript?

The relation between climate sensitivity and glacier compensation effect is also not clear and requires more explanation.
In the last sentence 'We also attempt to do this and that (glacier compensation effect and peak water)' I miss reasoning on why these attempts are needed.

[2] We thank the reviewer again for pointing out these shortcomings of the introduction sections. As mentioned in our reply [1], we shall rewrite the introduction section highlighting the need for studying the climate sensitivities of glacierised Himalayan catchments and the related research gap that exists.

For the studied catchments 1°C temperature rise by 2065 (RCP 2.6) is expected (Fig. S6). In Chandra (upper Dudhkoshi), the maximum positive temperature anomaly was 1.2°C (0.6°C). Therefore, the derived climate sensitivities may be used to understand the peak water - particularly in Chandra. We shall discuss these points in the revised manuscript, and indicate in Fig 7 and S6 what was the range of calibration, and where extrapolation was used.

Our discussions of glacier-compensation and peak-water effects were intended to demonstrate that the climate sensitivities obtained from the interannual variability of P, T and Q, can approximately capture some of the most well-known characteristics of the runoff of glacier-fed rivers. Clearly, a full-blown glacio-hydrological model may provide more accurate quantification of these effects, but the climate sensitivities provide useful insights into the mechanisms. To our knowledge, the role of climate sensitivities in controlling the variability of catchment runoff over different times scales have not been discussed in the literature.

Methodology
The workflow in this manuscript is not completely clear to me. The aim of the study is to assess climate sensitivities, because those can help to understand the variability and changes of streamflow. Since there is only limited streamflow data available, timeseries of streamflow are simulated with the VIC model. However, in theory, such models can also provide information on variability and change, so as a reader I need some argumentation why climate sensitivities are a useful alternative route, especially when there are no or only few streamflow observations available.

[3] Climate sensitivities are useful as they allow efficient quantification of runoff response, are useful in identifying possible biases in sophisticated high-complexity models, and can be exploited to gain simple intuitive understanding of the system response that often is not possible while using complex models. This motivates the present study of climate sensitivities derived from VIC model simulations. Please see reply [1] & [2] for further discussions.

The lack of streamflow data is a limitation for both process-based complex models, and a climate-sensitivity based approach, as both model parameters and sensitivities may require catchment-specific calibration. Studying the climate sensitivity of runoff, even if using glacio-hydrological models outputs, are useful as it may reveal some general characteristics of the climate sensitivities, and ensure some degree of transferability of parameters across catchments.

The simulations of streamflow are crucial here for the derivation of climate sensitivities, and I am surprised by the similar sensitivities of the two catchments, while their precipitation seasonality and mass balance type are so different.

Please see reply [23].

How is snowmelt simulated in the VIC model? Is there a different parameter for snow and glacier melt? If there is snow falling on the summer accumulation type glacier, is melt then also reduced in the model (albedo effect)?

[4] VIC model uses a two-layered snowpack, computing all the relevant energy fluxes, and using an energy-balance approach to obtain the snow melt. It uses a surface albedo paramterisation to incorporate the effects of snowfall and ageing of snow (Andreadis et al., 2009).

In the present study, we used VIC to estimate snow melt (including the albedo effect) over the off-glacier area and the snow-covered parts of the glacier. When melting of snow exposes glacial ice, a standard temperature-index model (Hock, 2003) is used to get the ice melt. This ice-melt module was added to VIC following several existing glacio-hydrological studies in the region using the same model (eg, Zhang et al, 2020, 2013, Tong et al, 2020, Chandel and Ghosh, 2021). Please see reply [17] for the overall model structure.

Have you tested if there is a difference in summer runoff sensitivity to summer precipitation in the non-glacierized parts?

[5] The table below shows climate sensitivities of summer runoff from off-glacier areas to summer precipitation ($S_{T/P}^{(r)}$) and annual precipitation($S_{T/P}^{(r)}$). The corresponding sensitivities to summer temperatures are also included.

| Catchment | $S_T^{(r)}$ ($S_T^{(r)}$) (mm yr$^{-1}$ $^o$C$^{-1}$) | $S_{P19}^{(r)}$ ($S_P^{(r)}$) (mm yr$^{-1}$ mm$^{-1}$) |
|---|---|---|
| Chandra | 22 ± 18 (21±13) | 0.28 ± 0.10 (0.56±0.04) |
| Upper Dushkoshi | 16 ± 14 (28±39) | 0.48 ± 0.07 (0.59±0.07) |

In summer monsoon dominated Dudhkoshi $S_P^{(r)}$ and $S_P^{(r)}$ are similar, but in winter snow-fed Chandra $S_T^{(r)}$ is considerably smaller.

How is ET modelled? This should be important for the non-glacierized runoff sensitivity to temperature.

[6] VIC uses the standard Penman-Monteith method (Monteith, 1965) to calculate the evapotranspiration. It considers different types of vegetation cover and tracks the corresponding transpiration, canopy evaporation, and the bare soil evaporation to compute the total ET (Liang et al. 1994). The mean ET loss for the catchments is about 30% of the mean precipitation (supplementary fig S5). In Dudhkoshi, ET variation is insignificant (small) with temperature (precipitation). In Chandra, it is the other way round. This suggests a water-limited condition in the summer monsoon-fed Dudhkoshi, and an energy-limited condition in the winter snow-fed Chandra, respectively (Please refer to the figures in our reply to the comment by Koji Fujita).

Regarding the parameter sensitivity tests, were the optimized DDF and ap parameters fixed? Low parameter sensitivity may suggest that the model is not very suitable to model the system.

[7] We have fixed 11 model parameters values as given in supplementary Table S2, and then optimised for DDF and $\alpha_P$. Subsequently, the sensitivity of summer runoff to these 11 parameters were computed at the optimal values of DDF and $\alpha_P$.
A low parameter sensitivity does not necessarily mean the corresponding parameter or processes are irrelevant. It implies the catchment-wide mean summer runoff is relatively robust to the uncertainties in these 11 parameters. The same parameters are important over shorter time and space scales for example.

Also, summer runoff may not be the optimal variable to test with, as timing of melt and snow/rain ratio will be important to model right to extract the sensitivities in a meaningful way.

[8] The summer season runoff, defined as the May to September runoff, is more than 80% of the annual runoff, and thus is a quantity of interest. Moreover, it coincides with the ablation season of Himalayan glaciers, so that they contribute significantly to the total summer runoff. These were the reasons we selected summer runoff.
We calibrated the models using available summer runoff and annual glacier mass balance data to ensure that our model captured these processes reasonably well. Snow/ice melt, and snow/rain ratio have to be right not only to get the summer runoff, but also the total and seasonal glacier mass balance.

Could Qg and Qr for the 40 year of simulations be easily plotted, and compared with other modelling studies?

[9] We are not aware of any available model results of glacier and off glacier runoff records for these two catchments. We could only compare our mean summer runoff with corresponding available estimates for the whole Dudhkoshi catchment (Nepal, 2016; Nepal 2017), Periche sub-catchment (Line 280--285 of the manuscript). We shall provide the modelled time-series of summer runoff, and its components in the supplementary.

Climate sensitivities are derived for catchment runoff, glacier runoff and non-glacierized runoff. There is a formula given (eq4) for how to derive catchment runoff sensitivity from the glacier and non-glacier runoff, but, if I am right, it is not used for the results. Has this been tested for?

[10] The climate sensitivities of catchment runoff ($S_{T/P}$) , glacier-runoff ($S_{T/P}^{(g)}$), and off-glacier runoff ($S_{T/P}^{(r)}$) are consistent using Eqs. (4-6) (given below).
$\delta Q = x\, \delta Q^{(g)} + (1 - x)\, \delta Q^{(r)}$ ------ (4)
$S_T = x\, S_T^{(g)} + (1-x)\, S_T^{(r)}$ ---------(5)
$S_P = x\, S_P^{(g)} + (1-x)\, S_P^{(r)}$ ---------(6)
Our reported sensitivity values are given in the table below, along with the corresponding values of $S_{T/P}$ computed with eqs 5-6 in (blue).

| Catchment | Glacier fraction (x) | $S_T$ (mm yr⁻¹ °C⁻¹) | $S_P$ (mm yr⁻¹ mm⁻¹) | $S_T^{(g)}$ (mm yr⁻¹ °C⁻¹) | $S_T^{(r)}$ (mm yr⁻¹ °C⁻¹) | $S_P^{(g)}$ (mm yr⁻¹ mm⁻¹) | $S_P^{(r)}$ (mm yr⁻¹ mm⁻¹) |
|---|---|---|---|---|---|---|---|
| Chandra | 0.25 | 117 (117.1) | 0.39 (0.39) | 405 | 21 | -0.12 | 0.56 |
| Upper Dudhkoshi | 0.20 | 116 (116.1) | 0.47 (0.47) | 469 | 28 | 0.00 | 0.59 |

In Eq. 8, the changes in runoff due to changes in glacier cover are estimated by the recent difference in runoff from the glaciers and the non-glacierized parts. This, however, neglects the process of usually increasing precipitation with elevation. For large changes of glacier cover this may become quite relevant.

[11] In the present formulation of eq 8, we ignored the spatial variation of runoff contributions within the off-glacier or glacierised areas as ERA5 annual precipitation had a relatively little (low) variability for Chandra (Dudhkoshi) as shown in the figure below.

[Figure]

Eq 8 can be modified if these variations were to be important, and we shall mention this point in the revised text. We emphasise that a monotonic elevation-dependent increase of precipitation does not capture the complex

spatial patterns of precipitation (Bookhagen & Burbank, 2010) in the rugged Himalayan landscapes. For example, an assumed uniform precipitation gradient with elevation likely led to an unrealistic east-west gradient in the modelled mass loss pattern in Chandra (Fig 3, Tawde et al., Annals of Glaciology, 2017).

Also, assuming 'the recent ratio of winter to summer runoff remain unchanged' contradicts many previous studies of increased winter flow and decreased summer flow. If these assumptions need to be made, I wonder how the results could be used, as many of the models do actually include these kinds of feedbacks.

[12] The point is well taken and we shall discuss this issue in the revised version. However, note that the above assumption was only used to estimate summer runoff from previously reported annual runoff at ~2050 available. These estimates were used for a general comparison with that reported in this study. In our simulations or analysis, we do not make any assumptions about this ratio. Incidentally, there were no significant changes in this ratio over 1980-2018 beyond the level of the interannual variability. Some previous studies in the Himalaya have also reported relatively small changes in the ratio by ~2050 (eg Lutz et al, 2016; Massood et al., 2015; Ragettli et al., 2016), though much larger changes are to be expected by the end of the century.

[Figure]

Throughout the results section, the climate sensitivities are presented as mm change per change in degree C or per change of mm of precipitation. Based on these outcomes, some sensitivities are regarded as zero. However, these results are misleading if they are not communicated in how much T and P varies per year. In general, it would be helpful, I think, to communicate them in percentage from the mean flow, and also present all of them in an overview table.

[13] As the studied catchments have similar summer runoff, absolute sensitivity values may be compared. However, we do agree that the percentage sensitivities are more useful, particularly for a comparison among catchments with a wide range of mean runoff. The percentage values are given in the table below, and these values do support our claim. Also, we note that in terms of percentages, the sensitivities in both the catchments are longer the same. We shall include these points in the revised version.

**Catchment summer runoff sensitivities in %-age**

| Catchment name | $S_T$ (%-age Q change per °C) | $S_P$ ( %-age Q change per 10% change in P) |
|---|---|---|
| Chandra | 11 ± 1 | 6 ± 1 |

| Upper Dushkoshi | 14 ± 4 | 9 ± 1 |
| --- | --- | --- |

**Glacier and off-glacier summer runoff sensitivities**

| Catchment name | $S_T^{(g)}$ (% of Q change per 1K warming) | $S_P^{(g)}$ (% of Q change due to 10% change in P) | $S_T^{(r)}$ (% of Q change per 1K warming) | $S_P^{(r)}$ (% of Q change due to 10% change in P) |
| --- | --- | --- | --- | --- |
| Chandra | 37 ± 2 | -2 ± 1 | 2 ± 1 | 9 ± 1 |
| Upper Dushkoshi | 58 ± 7 | 0 ± 0 | 3 ± 5 | 9 ± 1 |

Unclear descriptions
Throughout the manuscript there are quite some words missing or misspelled (please carefully check!), and unclear descriptions or presentation (see also list below). For example, units are missing in equations, and there is a mix of units in m and in mm.

[14] Thanks for pointing out these errors, and inconsistencies. We shall rectify them in the revised version. We shall address the specific issues in the revised version (see replies [1], [2], [15], [17], [18]).

The bias correction methods description is very unclear. Apparently, temperature is corrected based on station data, but precipitation not and instead is corrected via a calibration parameter. Why is that?

[15] The precipitation biases over the rugged Himalayan catchments (~1000 km$^2$) cannot be corrected using data from a single station because of a high spatial variability and a small correlation length of precipitation. Temperature has a longer correlation length and the data from a single station can be used. For example, in Dudhkoshi the ERA5 grid containing the station and the grid farthest off explained 85 and 79% of the variability of 15-day mean temperature. However, the corresponding explained variances were 72% and 55% for 15-day mean precipitation.

    Reanalysis data of temperature is expected to be more reliable than precipitation over the Himalaya, and the scale factor used for precipitation was necessary to satisfy water balance. It was calibrated here using the observed runoff and glacier mass balance which are basin-wide quantities, and we avoided using point-scale precipitation observation for this purpose as discussed above.

    We shall revise sect. 3.2.2 to clarify the rationale in a better way.

How is the meteorological input data used in VIC? Are T and P lapse rates used? If so, how are they obtained?

[16] VIC was run using the ERA5 hourly air-temperature, precipitation, and wind speed data (L158-162). Monthly mean temperature lapse rates were used to compute temperature for different elevation bands (supplementary Fig S2). No precipitation lapse rate was used (please see reply [11] for further arguments).

For the VIC modelling, how does the coupling between the glacierized and non-glacierized parts work? Is there snow redistribution from the non-glacierized parts onto the glacier? And does glacier melt contribute to baseflow?

[17] We ran the VIC model separately for glacerised and non-glacerised parts of the catchments. For each grid point, the streamflow contributions of the glacerised part (snowmelt, ice melt, rain), and nonglacierised parts (surface and subsurface runoff) are computed separately and fed into the routing model.

    Our present model did not consider snow redistribution from glacerised to non-glacierised parts of the catchment via avalanching (or wind redistribution) as acknowledged in present manuscript (L211). We do not consider any baseflow contribution from the glacier parts.

    We shall revise the model description to clarify these points. We also propose to add the following chart to describe the workflow better.

[Figure]

How are the mass balances calculated? Per catchment or per glacier? And how are they compared with available data, per glacier or per catchment?

[18] Mean glacier mass balance of the catchments were computed as the difference between the total snowfall and the total ice+snow melt over the total glacierised area in a catchment. These estimates with the corresponding available (~10) datasets catchment-wide geodetic mass balance that are listed in table S3. We shall revise sect 3.2 to clarify these points.

How is the glacier runoff modelled in a similar way to Huss and Hock?
Yes, we have modelled the glacier runoff in a similar way as Huss and Hock, 2018, described in Line 132--140 of the manuscript.

Are the same sensitivities as for the non-glacierized parts used for the parts that get de-glacierized?
[19] Yes. Please see reply [11] for further discussions.

In general, there are a lot of references to supplementary material, and I would suggest to better describe some of these in the main text.
[20] We shall revisit the supplementary reference list, and move the references that are better suited for the main text.

Lack of discussion
Section 4.9 is quite a deception to read. Basically, it summarizes the methods to obtain streamflow simulations. The approach used in this manuscript is very theoretical, and at least in the discussion section I think a translation again to the glacio-hydrological processes is needed (e.g. compensation of the glacierized and non-glacierized

runoff parts, connecting precipitation importance for mass balance to changes in summer streamflow, describing why temperature is not relevant for non-glacierized parts, interaction of P and T processes).

[21] We meant to describe in Sect 4.9 the limitation of the present study not that of the model. The model assumptions and limitations are already described in the methods section (which we shall revise/extend as described in our replies above). We shall rename sect 4.9 where describing in it the major limitations of our approach (limited field data, using model simulations to obtain the sensitivity, and the linear approximation implicit in the sensitivity approach).

We have provided physical explanations for and suggested mechanisms behind the precipitation insensitivity of glacier runoff (L342–L351), the temperature-insensitivity of off-glacier runoff (L356–L373), and also that of glacier-compensation (L409–L418) and peak water (L432–L443) effects. For the first two results we shall add additional discussions/graphs and revise our arguments (please see our reply to the comments by Koji Fujita). We took a theoretical approach towards the last two effects, as the corresponding physical mechanisms have been discussed in detail in the cited literature. Here we demonstrate that first-order quantitative explanations of these two effects follow just from the knowledge of the climate sensitivities.

Also, as mentioned before, it would be good to show what can be learned from these derived sensitivities, how can this approach be implemented to derive sensitivities in other catchments, or how do these results give a different perspective from what we already Know?
[22] Please see our replies earlier ([1], [2], and [3]).

And last but not least, I think a comparison with other climate sensitivities (also outside the Himalayas) is needed (e.g. He. 2021, Engelhardt et al., 2017, Moore et al., 2020), and some reasoning why there where no differences found between the two catchments (summer acc. types are thought to be very sensitive to temperature) and/or the differences in peak water timing in the two catchments, and the large differences in temperature sensitivities found in the studies that you cite in section 4.5
[23] We shall expand the comparative analysis of climates sensitivities of glacier runoff and glacier mass balance. We thank the reviewer for the suggested references, and shall include the compilation given below in the revised supplementary material.

**Climate sensitivity of catchment runoff**

| Catchment | $S_T$ (% of Q change per 1K warming) | $S_P$ ( % of Q change due to 10% change in P) | Reference |
|---|---|---|---|
| Engabreen | 24 | 2 | Engelhardt et al. (2015) |
| Ålfotbreen | 17 | 6 | Engelhardt et al, (2015) |
| Nigardsbreen | 21 | 4 | Engelhardt et al, (2015) |
| Storbreen | 19 | 3.3 | Engelhardt et al, (2015) |
| Ala-Archa (Northern Tienshan) | 9 | 7 | He (2021) |
| Dokriani catchment (Himalaya) | 20 | 16 | Azam and Srivastava, (2020) |
| Dudhkoshi basin (Himalaya) | 5 | 10 | Pokhrel et al, (2014) |
| Trambau Glacier basin | 27 | -0.6 | Fujita and Sakai, (2014) |
| **Chandra** (Himalaya) | **11 ± 1** | **6 ± 1** | **This study** |
| **Upper Dushkoshi** (Himalaya) | **14 ± 4** | **9 ± 1** | **This study** |

**Climate sensitivity of glacier runoff**

(sensitivity values are normalised by total catchment runoff (Q) as $Q^{(g)}$ vales are not available for all the studies)

| Catchment | $S_T^{(g)}$ (% of Q change per 1K warming) | $S_P^{(g)}$ (% of Q change due to 10% change in P) | Reference |
|---|---|---|---|
| Midtre Lovenbreen | 55 | 1 | Pramanik et al. (2018) |
| Kongsvegen | 71 | 3 | Pramanik et al. (2018) |
| Kronebreen-Holtedahlfonna | 55 | 4 | Pramanik et al. (2018) |
| Brewster glacier | 60 | 4 | Anderson et al. (2010) |
| La Paz, Bolivia | - | 6 | Soruco et al. (2015) |
| Trambau Glacier basin | 53 | -7 | Fujita and Sakai, (2014) |
| **Chandra** | **37 ± 2** | **-2 ± 1** | **This study** |
| **Upper Dushkoshi** | **58 ± 7** | **0 ± 0** | **This study** |

The above compilation indicates that the sensitivities reported by here are largely in line with those reported in the Himalaya and elsewhere. A weak precipitation sensitivity of glacier runoff in diverse climate settings is apparent from the above compilation, and  will be analysed/reported separately.

Note that the %-age sensitivities of the two studied catchments are not identical. Also the absolute values of sensitivities of glacier and off-glacier runoff, $(S_T^{(g)}, S_p^{(g)}, S_p$ etc) are significantly different (Table S5, S6), which shall be highlighted during revision.

Temperature is expected to have a stronger influence on mass balance on summer-accumulation type glaciers, due to conversion between snow and rain (Fujita, 2008; Kumar et al. 2019). However, the mass-balance sensitivity depends on other factors like glacier hypsometry. Therefore, a relatively higher temperature-sensitivity of glacier mass balance in summer-monsoon fed Dudhkoshi cannot be ruled out. Similar results were obtained using a detailed energy-balance model (see the figure reproduced from Sakai and Fujita (2017) with the catchments studied here marked). The other reported values from western and central Himalayan glaciers and catchments span a wide range (possibly

[Figure]

Mass balance sensitivity to temperature (m w.e. °C⁻¹)

reflecting the variability of climate and topography, underlying model assumptionz, model calibration, input data sets etc). Overall, the values presented are in the same range as previously known.

**Glacier mass balance sensitivity in the Himalaya**

| Catchment | temperature sensitivity (mm yr$^{-1}$ $^{o}$C$^{-1}$) | precipitation sensitivity (mm yr$^{-1}$ per 10% P change) | Reference |
|---|---|---|---|
| **Regional values** | | | |
| **Chandra** | **-475 ± 93** | **200 ± 42** | **This study** |
| Chandra | -570 to -640 | | Sakai & Fujita (2017); values extracted from supplementary figure S5 |
| Chandra | -160 | 90 | Tawde et al. (2017) |
| 4 western Himalayan glaciers | -240 to -835 | 60 to 90 | Wang et al. (2019) |
| Indus basin | -310 to -790 | | Shea and Immerzeel (2016) |
| **Upper Dushkoshi** | **-274 ± 46** | **50 ± 20** | **This study** |
| Dudhkoshi | -170 to -360 | | Sakai & Fujita (2017); values extracted from supplementary figure S5 |
| 5 Eastern/central Himalaya | -561 to -1000 | 50 to 80 | Wang et al. (2019) |
| Ganga basin | -290 to -760 | | Shea and Immerzeel (2016) |
| **Western Himalayan glaciers** | | | |
| Chhota Shigri glacier (Chandra) | -520 | 160 | Azam et al. (2014) |
| Naradu, Gara, Shaune Garang, Gor-Garang | -510, -310, -516, -240 | 74, 75, 72, 83 | Gaddam et al. (2017) |
| Shaune Garang, Gor-Garang, Gara, Siachen | -835, -709, -710, -240 | 60, 60, 60, 90 | Wang et al. (2019) |
| Siachen | -240 | 160 | Kumar et al. (2020) |
| Stok glacier | -320 | 120 | Soheb et al. (2020) |
| **Central/eastern Himalayan glaciers** | | | |
| AX010, Changmekhampu, Yala, Tipra | -1000, -656, -585, -561 | 80, 60, 50, 70 | Wang et al. (2019) |
| AX010 | -1000 | 153 | Kayastha et al. (1999) |
| Trambau | -900 -700 | 180 180 | Sunako et al. (2019) Kayastha et al. (1999) |
| Halji glacier | -1210 | 520 | Arndt et al. (2021) |

| Dokriani | -1110 | 240 | Azam and Srivastava (2020) |

Minor comments and technical corrections

Title: from – change to 'in'?
To be changed to OF in line with the text.
L3: catchments – change into glacierized catchments, also remove 'in order', and what is meant with 'the nature'?
L5: semi-distributeD
L22: response OF glacierized
L23: data – do you mean observations?
L27: also BE helpfulL37: An – a
L45: time series is – ARE
L54: is in – is LOCATED in
We accept these suggestions.

Table 1: do these values represent catchment mean? Or station values?
As stated in the caption these are catchment properties.

L60: solid to liquid or liquid to solid? Text and table say something different
It is the ratio of liquid to solid precipitation. We shall correct in L60.

Section 2 Study area – explain here the two glacier accumulation types
We shall discuss that in the revised version.

L65: bias corrected reanalysis data: bias corrected on what?
We shall refer the reader to the revised sect 3.2.2 here.

L67: relativvely – relatively
L75: concentrate ON
We accept these suggestions.

L84: please explain how you go from derivative in Q to anomaly Q
It follows from that definition, $Q(T,P) = Q_0(T_0,P_0) + \delta Q(P,T)$. We shall add this step in the revised version.

L97: onesdefined
We shall correct it.

L133: projected future changes in glacier area – How were they arrived? Is this data given per glacier? Or if per basin, do they match with the basins studied here?
L135: Also here, were these timeseries available for the catchment or for individual glaciers? How were they processed for this study?
The glacier fraction or temperature projections we used were for larger basins/regions, containing the two studied catchments. The regional values were used for the catchments without any modification.

L137: ignored – if precipitation changes were ignored, it means that in equation 15, change in P is zero and thus non-glacierized runoff is not included in the calculation for change in catchment runoff?
We followed the standard definition of peak water (Huss and Hock, 2018) as the change in runoff of the area that was glacierised at t=0. Here we set the origin at 2000 AD. As a result, only the first and the third terms on the RHS of eq 15 contribute. We shall clarify that in the revised text.

L138: gridded values available: please explain
L140 year 2002 and in L155 year 2002 – how did you derive glacier extent in 2000 then?
Glacier area is slow variable that does not change significantly in 2 years (see table 1).

L155-156: Please explain how the geodetic mass balances were obtained for the studied catchments
We shall add those details.

L185: size – elevation range?
We accept the suggestion.

L194: shrinkage of glacier fraction – is this value per decade? And is it the decline incatchment glacier cover or the decline in glacierized area?
These are decadal rates of changes in glacier area. These details are given in Table 1 which has been referred to.

L202-203: For melt calculations...... data set: very unclear, please rephrase
We shall revise it.

L236: j denoting individual records – what is meant here?
The sum is over the set 8 geodetic records or each of the catchments that are given in table S3. We shall revise the text and clarify.

L280: the sentences have a strange order here, with two times comparing to other studies
We shall revise it.

L295: th – the
We shall correct the error.

L305: Linear response: what is meant here?
We shall revise the title.

L321-L322: What about ET losses? Or change in storage
We thank the reviewer for pointing this out. In our reply to the comment by Koji Fujita, we showed that ~30% of the precipitation anomaly over the off-glacier part is lost as ET, and most of the excess precipitation over the glacierised parts falls as snow and contributes to glacier accumulation. We shall revise the text appropriately.

L338: has – have
We shall correct the error.

L354: where can I see this effect of stabilizing scaled with glacier fraction?
L390 and 399: to – two
We shall correct the error.

L392: accurate sensitivities – sounds plausible, but how to derive them?
Climate sensitivities may be obtained from observations whenever available (L30–L31) - then a model run can be avoided. When observations are not there, available model simulations have to be used - still useful when models have to be run for new scenarios and the model is computationally heavy. When neither observations nor model simulations are not available, climate sensitivities are not known. However, there may still be some general properties of climate sensitivity that may allow some insight about future changes. For example, some climate sensitivities could be small or negligible in general, e.g., an approximately precipitation insensitive glacier runoff - please see reply [23].

L402 and 472: depndence

We shall correct the error.

L406-407: Do they propose that in this paper?
Chen and Ohmura. (1990) used an empirical paramterisation of the glacier compensation curve, calibrated the same using data from some alpine catchments, and showed that it can explain the observed multidecadal runof changes in the catchments.

Section 4.8.3 – Why is peakwater not calculated for catchment runoff?
We followed the standard definition of peak water, eg, that of Huss and Hock, 2018.

Figure 1 – It may be an idea to indicate the sub-catchments which were you used in other studies and that you use for comparison of your results
We shall revise the figure.

Figure 2 – Please provide the meaning of the parameters
We shall add the information.

Figure 4 – what do the different dots represent?
The modelled and observed (geodetic) decadal mass balance for the two catchments. Each of these data points are also given in table S3.

**References**

Anderson, B., Mackintosh, A., Stumm, D., George, L., Kerr, T., Winter-Billington, A., & Fitzsimons, S. (2010). Climate sensitivity of a high-precipitation glacier in New Zealand. *Journal of Glaciology*, 56(195), 114-128.

Arndt, A., Scherer, D., & Schneider, C. (2021). Atmosphere Driven Mass-Balance Sensitivity of Halji Glacier, Himalayas. Atmosphere, 12(4), 426.

Azam, M. F., Kargel, J. S., Shea, J. M., Nepal, S., Haritashya, U. K., Srivastava, S., ... & Bahuguna, I. M. (2021). Glaciohydrology of the Himalaya-Karakoram. *Science*.

Azam, M. F., Wagnon, P., Vincent, C., Ramanathan, A., Linda, A., Singh, V. B. (2014). Reconstruction of the annual mass balance of Chhota Shigri glacier, Western Himalaya, India, since 1969. Annals of Glaciology, 55(66), 69-80.

Azam, M. F., and Srivastava, S. (2020). Mass balance and runoff modelling of partially debris-covered Dokriani Glacier in monsoon-dominated Himalaya using ERA5 data since 1979. Journal of Hydrology, 125432.

Banerjee, A., Patil, D., & Jadhav, A. (2020). Possible biases in scaling-based estimates of glacier change: a case study in the Himalaya. The Cryosphere, 14(9), 3235-3247.

Bookhagen, B., & Burbank, D. W. (2006). Topography, relief, and TRMM-derived rainfall variations along the Himalaya. Geophysical Research Letters, 33(8).

Chandel, V. S., & Ghosh, S. (2021). Components of Himalayan River Flows in a Changing Climate. Water Resources Research, 57(2), e2020WR027589.

Chen, J., and Ohmura, A. (1990). On the influence of Alpine glaciers on runoff. IAHS Publ, 193, 117-125.

Engelhardt, M., Schuler, T. V., & Andreassen, L. M. (2015). Sensitivities of glacier mass balance and runoff to climate perturbations in Norway. *Annals of Glaciology*, 56(70), 79-88.

Fujita, K. and Sakai, A.: Modelling runoff from a Himalayan debris-covered glacier, Hydrol. Earth Syst. Sci., 18, 2679–2694, https://doi.org/10.5194/hess-18-2679-2014, 2014

Gaddam, V. K., Kulkarni, A. V., & Gupta, A. K. (2017). Reconstruction of specific mass balance for glaciers in Western Himalaya using seasonal sensitivity characteristic (s). Journal of Earth System Science, 126(4), 1-10.

He, Z. (2021). Sensitivities of hydrological processes to climate changes in a Central Asian glacierized basin. *Frontiers in Water*, *3*, 46.

Kayastha RB, Ohata T, Ageta Y (1999). Application of a mass-balance model to a Himalayan glacier. Journal of Glaciology. 45(151):559-567. doi:10.3189/S002214300000143X.

Khadka, M., Kayastha, R. B., & Kayastha, R. (2020). Future projection of cryospheric and hydrologic regimes in Koshi River basin, Central Himalaya, using coupled glacier dynamics and glacio-hydrological models. *Journal of Glaciology*, *66*(259), 831-845.

Kumar, A., Negi, H. S., & Kumar, K. (2020). Long-term mass balance modelling (1986–2018) and climate sensitivity of Siachen Glacier, East Karakoram. Environmental monitoring and assessment, 192(6), 1-16.

Leclercq, P. W., Oerlemans, J., & Cogley, J. G. (2011). Estimating the glacier contribution to sea-level rise for the period 1800–2005. Surveys in Geophysics, 32(4-5), 519.

Lutz, A. F., Immerzeel, W. W., Kraaijenbrink, P. D., Shrestha, A. B., & Bierkens, M. F. (2016). Climate change impacts on the upper Indus hydrology: sources, shifts and extremes. *PloS one*, *11*(11), e0165630.

Masood, M., Yeh, P. F., Hanasaki, N., & Takeuchi, K. (2015). Model study of the impacts of future climate change on the hydrology of Ganges–Brahmaputra–Meghna basin. *Hydrology and Earth System Sciences*, *19*(2), 747-770.

Mölg, T., Maussion, F., Yang, W., Scherer, D. (2012). The footprint of Asian monsoon dynamics in the mass and energy balance of a Tibetan glacier. The Cryosphere, 6(6), 1445.

Oerlemans, J. (2005). Extracting a climate signal from 169 glacier records. Science, 308(5722), 675-677.

Pokhrel, B. K., Chevallier, P., Andreassian, V., Tahir, A. A., Arnaud, Y., Neppel, L., ... Budhathoki, K. P. (2014). Comparison of two snowmelt modelling approaches in the Dudh Koshi basin (eastern Himalayas, Nepal). Hydrological sciences journal, 59(8), 1507-1518.

Pramanik, A., Van Pelt, W., Kohler, J., & Schuler, T. V. (2018). Simulating climatic mass balance, seasonal snow development and associated freshwater runoff in the Kongsfjord basin, Svalbard (1980–2016). *Journal of Glaciology*, *64*(248), 943-956.

Ragettli, S., Immerzeel, W. W., & Pellicciotti, F. (2016). Contrasting climate change impact on river flows from high-altitude catchments in the Himalayan and Andes Mountains. Proceedings of the National Academy of Sciences, 113(33), 9222-9227.

Sakai, A., & Fujita, K. (2017). Contrasting glacier responses to recent climate change in high-mountain Asia. Scientific reports, 7(1), 1-8.

Shea, J. M., and Immerzeel, W. W. (2016). An assessment of basin-scale glaciological and hydrological sensitivities in the Hindu Kush–Himalaya. Annals of Glaciology, 57(71), 308-318.

Soheb, M., Ramanathan, A., Angchuk, T., Mandal, A., Kumar, N., & Lotus, S. (2020). Mass-balance observation, reconstruction and sensitivity of Stok glacier, Ladakh region, India, between 1978 and 2019. Journal of Glaciology, 66(258), 627-642.

Soruco, A., Vincent, C., Rabatel, A., Francou, B., Thibert, E., Sicart, J. E., & Condom, T. (2015). Contribution of glacier runoff to water resources of La Paz city, Bolivia (16 S). *Annals of Glaciology*, *56*(70), 147-154.

Sunako, S., Fujita, K., Sakai, A., & Kayastha, R. B. (2019). Mass balance of Trambau Glacier, Rolwaling region, Nepal Himalaya: in-situ observations, long-term reconstruction and mass-balance sensitivity. Journal of Glaciology, 65(252), 605-616.

Tawde, S., Kulkarni, A., & Bala, G. (2017). An estimate of glacier mass balance for the Chandra basin, western Himalaya, for the period 19842012. Annals of Glaciology, 58(75pt2), 99-109. doi:10.1017/aog.2017.18

Tong, K., Su, F., & Li, C. (2020). Modeling of water fluxes and budget in Nam Co Basin during 1979–2013. Journal of Hydrometeorology, 21(4), 829-844.

Wang, R., Liu, S., Shangguan, D., Radic, V., Zhang, Y. (2019). Spatial heterogeneity in glacier mass-balance sensitivity across High Mountain Asia. Water, 11(4), 776.

Zhang, L., Su, F., Yang, D., Hao, Z., & Tong, K. (2013). Discharge regime and simulation for the upstream of major rivers over Tibetan Plateau. Journal of Geophysical Research: Atmospheres, 118(15), 8500-8518.

Zhang, H., Li, Z., Zhou, P., Zhu, X., & Wang, L. (2018). Mass-balance observations and reconstruction for Haxilegen Glacier No. 51, eastern Tien Shan, from 1999 to 2015. Journal of Glaciology, 64(247), 689-699.

Zhang, Y., Xu, C. Y., Hao, Z., Zhang, L., Ju, Q., & Lai, X. (2020). Variation of Melt Water and Rainfall Runoff and Their Impacts on Streamflow Changes during Recent Decades in Two Tibetan Plateau Basins. Water, 12(11), 3112.

---

## Author Comment (AC3)

**Authors' response (AC3) to comments (CC1) by Koji Fujita**

**Comment:**
The model of this study yielded the insensitive response of glacier runoff to precipitation change but this is not discussed in depth. The model can resolve how each component responds (that's why we utilize numerical models, right?). Figure 11a of Fujita and Sakai (2014, HESS) could be helpful for this issue. Runoff responses to precipitation change are different in ice-containing and ice-free surfaces, and the compensation of these opposite responses could yield the insensitive response. I suggest that the authors perform this kind of analysis. It would be interesting if the authors find a different reason.

**Reply:**

Fujita & Sakai (2014) showed that for a glacier catchment in the central Himalaya, a contrasting response of runoff of the clean-ice ('glacier') and debris-covered ('debris') parts, and that of the off-glacier part ('ground') led to a weak response of the catchment runoff. For a 10% increase in precipitation, the runoff of the glacier+debris parts decreases by ~7% and that of the 'ground' increases by ~7%, leading to a subdued response of the total runoff (less than 1% change).

    What we are seeing in Chandra and Dudhkoshi, is a subdued response of the glacier runoff itself (0-1% change) for a 10% change in P. Our observations are in line with those reported for a few other glaciers in the world (see reply [23] in AC2).

    We thank Koji Fujita for his suggestions about plotting the response of individual runoff components. The plots are provided below. Overall, they indicate that precipitation anomalies on glaciers contribute to that of accumulation, so that glacier runoff remains largely unaffected. In contrast, a higher temperature causes a higher glacier melt, and thus, a glacier runoff.

**Sensitivities of summer runoff of the glacierised area**

The anomalies of glacier runoff ($Q^{(g)}$), and its components snowmelt (SM), glacier ice melt (GM), and rainfall (RF) for the glacierised parts of the catchments are shown below.

**Chandra:**

[Figure]

**Upper Dudhkoshi:**

[Figure]

In both the catchments, rising GM with mean summer temperature causes a high temperature sensitivity of $Q^{(g)}$ (410±28 and 435±72 mm yr$^{-1}$ °C$^{-1}$). RF and SM are largely unaffected by the changes with temperature. A weak (insignificant) decline in SM in Chandra with increasing summer temperature is likely due to a weak but insignificant anticorrelation (r = -0.3) between summer temperature and annual precipitation.

With increasing precipitation, RF remains unchanged and SM shows a very weak (Chandra) or no (Dudhkoshi) increase. Thus a higher precipitation contributes mostly to snow accumulation on glaciers. In addition, a higher snowcover and/or an association between higher-than-normal precipitation and less-than-normal temperature, causes a weak decline in GM, and consequently, in $Q^{(g)}$ (−0.12±0.08 and 0.00±0.02 mm yr$^{-1}$ mm$^{-1}$ ).

**Sensitivities of summer runoff of the non-glacierised area**

The anomalies of off-glacier runoff ($Q^{(r)}$), and its components, surface runoff (R) and groundwater/baseflow (G) are plotted below. The corresponding evapotranspiration (ET) anomalies are also shown.

**Chandra:**

[Figure]

**Upper Dudhkoshi:**

[Figure]

The above plots suggest a roughly equal distribution of any precipitation change into the corresponding changes in R, G, and ET ,such that ~60% of the additional precipitation contributes to runoff changes. Interestingly, ET anomalies in Chandra (Dudhkoshi) are controlled by the summer temperature (precipitation) suggesting energy- (water-) limited conditions.

Note that the sensitivities as obtained from in the above plots are slightly different than those quoted in the main text, as only the anomalies over the calibration period of 1997-2018 were used in the main text.

**References**

Fujita, K. and Sakai, A.: Modelling runoff from a Himalayan debris-covered glacier, Hydrol. Earth Syst. Sci., 18, 2679–2694, https://doi.org/10.5194/hess-18-2679-2014, 2014